# On the Emergence of Position Bias in Transformers

**Xinyi Wu** [1]   **Yifei Wang** [2]   **Stefanie Jegelka** [3] [2]   **Ali Jadbabaie** [1]

## Abstract

Recent studies have revealed various manifestations of position bias in transformer architectures, from the "lost-in-the-middle" phenomenon to attention sinks, yet a comprehensive theoretical understanding of how attention masks and positional encodings shape these biases remains elusive. This paper presents a graph-theoretic framework for analyzing position bias in multi-layer attention. Modeling attention masks as directed graphs, we quantify how tokens interact with contextual information based on their sequential positions. We uncover two key insights: First, causal masking inherently biases attention toward earlier positions, as tokens in deeper layers attend to increasingly more contextualized representations of earlier tokens. Second, we characterize the competing effects of the causal mask and relative positional encodings, such as the decay mask and rotary positional encoding (RoPE): while both mechanisms introduce distance-based decay within individual attention maps, their aggregate effect across multiple attention layers—coupled with the causal mask—leads to a trade-off between the long-term decay effects and the cumulative importance of early sequence positions. Through controlled numerical experiments, we not only validate our theoretical findings but also reproduce position biases observed in real-world LLMs. Our framework offers a principled foundation for understanding positional biases in transformers, shedding light on the complex interplay of attention mechanism components and guiding more informed architectural design.

## 1. Introduction

The attention mechanism is central to transformer architectures (Vaswani et al., 2017), which form the backbone of state-of-the-art foundation models, including large language models (LLMs). Its success lies in its ability to dynamically weigh input elements based on their relevance, enabling efficient handling of complex dependencies (Bahdanau et al., 2015; Kim et al., 2017). However, despite this widespread success, many questions remain unanswered regarding how these mechanisms process information and the artifacts they may introduce. Developing a deeper theoretical understanding of their inner workings is essential – not only to better interpret existing models but also to guide the design of more robust and powerful architectures.

One particularly intriguing aspect that demands such a theoretical investigation is *position bias*, i.e., the bias of the model to focus on certain regions of the input, which significantly impacts the performance and reliability of transformers and LLMs (Hou et al., 2024; Wang et al., 2024; Zheng et al., 2023). For instance, these models often suffer from the "lost-in-the-middle" problem, where retrieval accuracy significantly degrades for information positioned in the middle of the input sequence compared to information at the beginning or end (Guo & Vosoughi, 2024; Liu et al., 2024; Zhang et al., 2024). Similarly, in-context learning is highly sensitive to the order of illustrative examples: simply shuffling independently and identically distributed (i.i.d.) examples can lead to significant performance degradation (Lu et al., 2022; Min et al., 2022; Zhao et al., 2021). Moreover, recent research has also revealed that attention sinks (Gu et al., 2025; Guo et al., 2024; Xiao et al., 2024) – positions that attract disproportionately high attention weights – arise at certain positions regardless of semantic relevance, suggesting an inherent positional bias.

These empirical findings suggest that while transformers effectively encode and process positional information through the combined use of attention masks and positional encodings (PEs) (Fang et al., 2025; Wang et al., 2024), these design elements also appear to introduce systematic positional biases, often independent of semantic content. This raises a fundamental and intriguing question about the role of positional information in attention mechanisms:

> *How do attention masks and positional encodings shape position bias in transformers?*

To address the question, we develop a graph-theoretic framework for analyzing attention score distributions in multi-layer attention settings. Related to the empirical tool of attention rollout (Abnar & Zuidema, 2020) and building

---

[1]MIT IDSS & LIDS [2]MIT CSAIL [3]TU Munich. Correspondence to: Xinyi Wu <xinyiwu@mit.edu>.

*Proceedings of the 42nd International Conference on Machine Learning*, Vancouver, Canada. PMLR 267, 2025. Copyright 2025 by the author(s).

Table 1: Summary of our results and their connections to empirical observations on position bias reported in the literature.

| Empirical Observations on Position Bias | Our Results |
| --- | --- |
| Positional information induced by the causal mask (Barbero et al., 2024a; Kazemnejad et al., 2023; Wang et al., 2024) | Theorem 4.1, Section 5.2 |
| Decay effects induced by relative PEs (Su et al., 2023) | Lemma 4.4-4.6, Section 5.1 |
| Interplay between the causal mask and relative PEs (Wang et al., 2024) | Theorem 4.5-4.7, Section 5.1 |
| Attention sinks (Gu et al., 2025; Xiao et al., 2024) | Theorem 4.1-4.3, Appendix K.4 |
| The "lost-in-the-middle" phenomenon (Guo & Vosoughi, 2024; Liu et al., 2024) | Section 5.2 |

upon the theoretical analysis of attention from a graph perspective (Barbero et al., 2024a; Wu et al., 2023; 2024), we model attention masks as directed graphs, enabling rigorous mathematical analysis of attention patterns. This approach proves particularly powerful for studying multi-layer attention mechanisms, as it allows us to precisely quantify how each token's contextual representation is composed from information at different positions in the sequence. By tracking the information flow through the attention layers, we can systematically examine how positional biases emerge and propagate across layers, providing insights into the complex interplay between attention masks, PEs, and the network's depth.

**Our contributions are summarized as follows:**

- We develop a graph-theoretic framework that unifies and advances understanding of position bias in transformers, offering deeper insights into diverse empirical observations documented in the literature (Table 1).
- We show that causal masking in transformers inherently biases attention toward earlier positions in deep networks. This happens as tokens in deeper layers attend to increasingly more contextualized representations of earlier tokens, thereby amplifying the influence of initial positions. We derive analogous results for the sliding-window mask and the prefix mask, highlighting the generalizability of our framework.
- We uncover a nuanced interaction between causal masking and relative PEs, such as decay masks and rotary positional encoding (RoPE). Our findings highlight a trade-off in multi-layer attention networks, where local decay effects within individual layers are counterbalanced by the cumulative importance of early sequence positions. These results provide a deeper understanding of how PE and masking interact in deep attention-based architectures, with design implications about how to balance local and global context.
- We support our theoretical findings with experiments, empirically validating that deeper attention layers amplify the bias toward earlier parts of the sequence, while relative PEs partially mitigate this effect. Through carefully controlled numerical experiments, we further investigate the role of data in shaping position bias and how causal masking implicitly leverages positional information.

## 2. Related Work

**Position bias in transformers** Position bias in transformer models has emerged as a critical challenge across diverse applications. In information retrieval and ranking, Guo & Vosoughi (2024); Hou et al. (2024); Liu et al. (2024); Zheng et al. (2023) demonstrated systematic degradation of performance due to positional dependencies. Similarly, in in-context learning, model performance can vary dramatically based solely on the order of examples (Fang et al., 2025; Lu et al., 2022; Min et al., 2022; Zhao et al., 2021). While mitigation strategies such as novel PEs (Kazemnejad et al., 2023; Zhang et al., 2024), alternative masking techniques (Fang et al., 2025; Wang et al., 2024) and bootstrapping (Hou et al., 2024) have been proposed, they remain task-specific and empirically driven. This gap between empirical observations and theoretical understanding highlights the need for a rigorous analysis of how transformers process and integrate positional information through attention.

**The effect of attention masks and PEs in transformers** The role of attention masks and PEs in transformers has been explored from various perspectives. Yun et al. (2020b) analyzed the function approximation power of transformers under different masking schemes, while Wu et al. (2023; 2024) investigated the role of attention masks in mitigating rank collapse. Moreover, Gu et al. (2025) empirically examined how attention masks affect the emergence of attention sinks. As for PEs, Kazemnejad et al. (2023) studied their role in length generalization, and Barbero et al. (2024b) analyzed RoPE's use of feature dimensions. Additionally, Wang et al. (2024) empirically observed that both causal masking and RoPE introduce position dependencies in LLMs. Despite these advances, fundamental questions remain about the mechanisms through which attention masks and PEs enable transformers to process and integrate positional information, as well as the nature of the systematic positional biases that emerge as a result.

## 3. Problem Setup

**Notation** We use the shorthand $[n] := \{1, \dots, n\}$. For a matrix $M$, we denote its $i$-th row by $M_{i,:}$ and its $j$-th column by $M_{:,j}$. Throughout the analysis in the paper, we formalize the attention mask to be a directed graph $\mathcal{G}$. Formally, we represent a directed graph with $N$ nodes by $\mathcal{G}$

and let $E(\mathcal{G})$ be the set of directed edges of $\mathcal{G}$. A directed edge $(j, i) \in E(\mathcal{G})$ from node $j$ to $i$ in $\mathcal{G}$ means that in the attention mechanism, token $j$ serves as a direct context for token $i$ or token $i$ attends to token $j$. The set $\mathcal{N}_i$ of all neighbors of node $i$ is then $\{k : (k, i) \in E(\mathcal{G})\}$. We say a node $v$ is *reachable* from node $u$ in a directed graph $\mathcal{G}$ if there is a directed path $(u, n_1), (n_1, n_2), ..., (n_k, v)$ from $u$ to $v$. In the attention mechanism, this means that token $u$ serves as a direct or indirect context for token $v$.

Furthermore, we will be using the following graph-theoretic terminology (see Figure 1 for a schematic illustration):

**Definition 3.1** (Center Node). A node $v$ from which every node in the directed graph $\mathcal{G}$ is reachable is called a *center node*.

### 3.1. (Masked) Attention Mechanism

Given the representation $X \in \mathbb{R}^{N \times d}$ of $N$ tokens, the raw attention score matrix is computed as $Z = XW_Q(XW_K)^\top / \sqrt{d_{QK}}$, where $W_Q, W_K \in \mathbb{R}^{d \times d'}$ are the query and the key matrix, respectively, and $\sqrt{d_{QK}}$ is a temperature term to control the scale of raw attention scores. Without loss of generality, we assume $d_{QK} = 1$ in our analysis. To enforce a masked attention, we create a sparse attention matrix $A \in \mathbb{R}^{N \times N}$ based on $Z$ whose sparsity pattern is specified by a directed graph $\mathcal{G}$: we normalize $Z_{ij}$ among all allowed token attention interactions $(k, i) \in E(\mathcal{G})$ such that if $(j, i) \in E(\mathcal{G})$,

$$A_{ij} = \text{softmax}_{\mathcal{G}}(Z_{ij}) = \frac{\exp(Z_{ij})}{\sum_{k \in \mathcal{N}_i} \exp(Z_{ik})} \;,$$

and $A_{ij} = 0$ otherwise.

### 3.2. Attention Update

For our analysis, we consider single-head (masked) self-attention networks (SANs). The layerwise update rule can be written as

$$A^{(t)} = \text{softmax}_{\mathcal{G}^{(t)}} \left( X^{(t)} W_Q^{(t)} (X^{(t)} W_K^{(t)})^\top / \sqrt{d_{QK}} \right)$$

$$X^{(t+1)} = A^{(t)} X^{(t)} W_V^{(t)} , \tag{1}$$

where $W_V^{(t)} \in \mathbb{R}^{d \times d'}$ is the value matrix. For simplicity, throughout the paper, we assume that $d = d'$ and $\mathcal{G}^{(t)} = \mathcal{G}$. Yet the results can be easily generalized to the case where masks are time-varying and satisfy regularity conditions.

### 3.3. Relative Positional Encoding

**Decay Mask**  The decay mask represents the relative distance between two tokens by introducing an explicit bias favoring more recent tokens. Formally, it can be written as:

$$D_{ij} = \begin{cases} -(i - j)m & \text{if } j \leq i \\ 0 & \text{otherwise} . \end{cases}$$

Then applying the decay mask is essentially

$$A_{\text{decay}}^{(t)} = \text{softmax}_{\mathcal{G}}(X^{(t)} W_Q^{(t)} (X^{(t)} W_K^{(t)})^\top + D) . \tag{2}$$

Note that while the decay mask formulation follows ALiBi (Press et al., 2022), it can be generalized to more complex variants such as KERPLE (Chi et al., 2022).

**Rotary Positional Encoding (RoPE)**  Another way to encode the relative positional information is through RoPE (Su et al., 2023), which applies a rotation to query and key embeddings by an angle proportional to the token's position index within the sequence. Formally, the rotation operation applied to each query or key $X_{i,:}W_{\{Q,K\}}$ can be written as

$$(\hat{X}_{\{Q,K\}})_{i,:} = X_{i,:}W_{\{Q,K\}}R_{\Theta,i}^d \tag{3}$$

where

$$R_{\Theta,i}^d = \begin{bmatrix} \cos i\theta_1 & \sin i\theta_1 & 0 & 0 & \cdots & 0 & 0 \\ -\sin i\theta_1 & \cos i\theta_1 & 0 & 0 & \cdots & 0 & 0 \\ 0 & 0 & \cos i\theta_2 & \sin i\theta_2 & \cdots & 0 & 0 \\ 0 & 0 & -\sin i\theta_2 & \cos i\theta_2 & \cdots & 0 & 0 \\ \vdots & \vdots & \vdots & \vdots & \ddots & \vdots & \vdots \\ 0 & 0 & 0 & 0 & \cdots & \cos i\theta_{d/2} & \sin i\theta_{d/2} \\ 0 & 0 & 0 & 0 & \cdots & -\sin i\theta_{d/2} & \cos i\theta_{d/2} \end{bmatrix}$$

is the rotation matrix with a set of pre-defined base rotational angles $\Theta = \{0 \leq \theta_1 \leq \cdots \leq \theta_{d/2}\}$. Then the raw attention scores under RoPE $Z_{\text{RoPE}}$ become

$$(Z_{\text{RoPE}})_{ij} = (X_{i,:}W_Q R_{\theta,i}^d)(X_{j,:}W_K R_{\theta,j}^d)^\top$$
$$= X_{i,:}W_Q R_{\theta,i-j}^d W_K^\top X_{j,:}^\top,$$

which distorts the original raw attention scores based on the relative token distances. The final attention scores under RoPE are calculated as $A_{\text{RoPE}}^{(t)} = \text{softmax}_{\mathcal{G}} \left( Z_{\text{RoPE}}^{(t)} \right)$.

## 4. Main Results

In the transformer model, the attention mechanism is the sole module that allows tokens to interact with one another and incorporate contextual information from the sequence. It iteratively refines the contextual representation of each token across layers, allowing information to flow and accumulate based on relevance. This concept of contextualization through attention has its origins in the development of attention mechanisms, which predate transformers (Bahdanau et al., 2015; Kim et al., 2017). From the perspective of contextualization, the attention mechanism can be expressed in the following form (Kim et al., 2017):

$$X_{i,:}^{(t+1)} = \sum_{j=1}^{N} \underbrace{(A^{(t)} \cdots A^{(0)})_{ij}}_{\mathbb{P}^{(t)}(z_i = j | X^{(0)})} \cdot \underbrace{X_{j,:}^{(0)} W_V^{(0)} \cdots W_V^{(t)}}_{f^{(t)}(X_{z_i,:}^{(0)})}, \tag{4}$$

where $z_i$ is a categorical latent variable with a sample space $\{1, \ldots, N\}$ that selects the input $X_{j,:}$ to provide context for token $i$. In this formulation, $A^{(t)}$ represents the attention

matrix at layer $t$, $\mathbb{P}^{(t)}(z_i = j \mid X^{(0)})$ denotes the cumulative probability of selecting input token $j$ as the context for token $i$ at depth $t$, and $f^{(t)}(\cdot)$ is a learned transformation function.

This probabilistic formulation reveals two key aspects of the attention mechanism: it acts as both a context selector and a feature aggregator. As a selector, it assigns probabilities $\mathbb{P}^{(t)}$ that quantify the relevance of each token $j$ to target token $i$ at depth $t$. As an aggregator, it combines these selected contexts weighted by their respective probabilities $\mathbb{P}^{(t)}$ to form the contextualized representation $X^{(t)}$. Since position bias fundamentally manifests as systematic preferences in how tokens select and incorporate context from different positions, analyzing the attention mechanism's behavior is crucial for understanding these biases. By examining how attention masks and PEs affect the probability distribution $\mathbb{P}^{(t)}$, we can investigate how position-dependent patterns emerge and propagate through multi-layer attention in transformers.

**Attention rollout** The quantity analyzed in our framework, $\mathbb{P}^{(t)}(z_i = j \mid X^{(0)})$, coincides with the attention rollout metric proposed by Abnar & Zuidema (2020). However, our work was developed independently, and the motivation is fundamentally different: whereas attention rollout serves as an empirical visualization tool applied to specific input sequences, we treat the same quantity as a theoretical object and analyze it across arbitrary inputs to reveal the model's inductive biases.

Finally, we adopt the following assumptions in our analysis:

**A1** There exists $C \in \mathbb{R}$ such that

$$\max_{t \in \mathbb{N}}\{\|W_Q^{(t)}\|_2, \|W_K^{(t)}\|_2\} \leq C.$$

**A2** The sequence $\left\{\|\prod_{t=0}^k W_V^{(t)}\|_2\right\}_{k=0}^{\infty}$ is bounded.

**A1** assumes that the key and query weight matrices are bounded, which is crucial for efficient attention computation in practice (Alman & Song, 2023), whereas **A2** is to ensure boundedness of the node representations' trajectories $X^{(t)}$ for all $t \geq 0$ (Wu et al., 2024).

### 4.1. Attention Masks: A Graph-Theoretic View

We first analyze the case without PEs, focusing on the effect of attention masks. A graph-theoretic perspective offers a powerful framework for analyze multi-layer attention: the flow of attention across tokens can be represented as paths in a directed graph defined by the mask, where each path captures how information is transmitted between tokens (see Figure 1 for an illustration). The number of steps in a path corresponds to the number of layers. By accounting for all such paths, we can quantify the cumulative influence of each token in the context computation of other tokens.

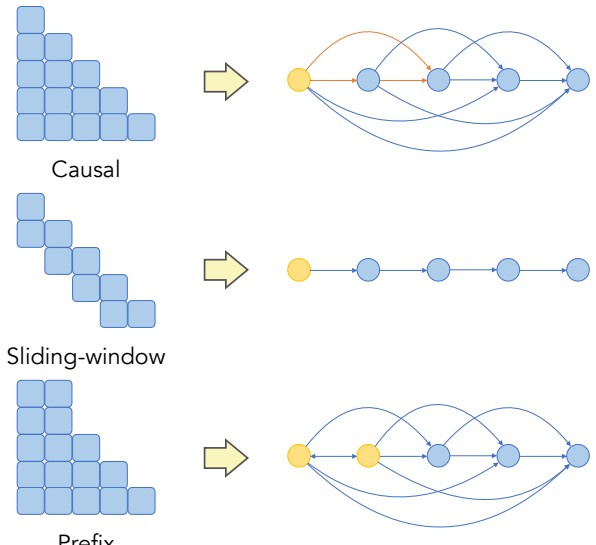

Causal

Sliding-window

Prefix

Figure 1: Three types of attention masks and their corresponding directed graphs $\mathcal{G}$ used in the analysis (self-loops are omitted for clarity). A directed edge from token $j$ to $i$ indicates that $i$ attends to $j$. The center node(s) (Definition 3.1), highlighted in yellow, represent tokens that can be directly or indirectly attended to by all other tokens in the sequence. As depicted in the top row, the graph-theoretic formulation captures both direct and indirect contributions of tokens to the overall context, providing a comprehensive view of the token interactions under multi-layer attention.

Through the graph-theoretic view, our first result states that for a causal mask $\mathcal{G}$, as tokens in deeper layers attend to increasingly more contextualized representations of earlier tokens, the context of each token converges exponentially toward the first token in the sequence.

**Theorem 4.1.** *Let $\mathcal{G}$ be the causal mask. Under **A1**-**A2**, given $X^{(0)} \in \mathbb{R}^{N \times d}$, for every token $i \in [N]$,*

$$\lim_{t \to \infty} \mathbb{P}^{(t)}(z_i = 1 | X^{(0)}) = 1.$$

*Moreover, there exist $0 < C, \epsilon < 1$ where $N\epsilon < 1$ such that*

$$\mathbb{P}^{(t)}(z_i = j | X^{(0)}) \leq C(1 - (j-1)\epsilon)^t.$$

*for all $1 < j \leq i$ and $t \geq 0$.*

Theorem 4.1 reveals that in multi-layer causal attention, positional bias toward earlier sequence positions intensifies with depth – regardless of semantic content. This phenomenon arises from the nature of multi-layer attention: starting from the second layer, tokens no longer attend to raw inputs but instead to contextualized tokens, i.e., representations transformed by prior attention layers. Combined with the sequential structure of the causal mask, this iterative process amplifies the role of earlier tokens, as they influence later ones not only as direct context but also indirectly through intermediate tokens along the path. We discuss a few intriguing implications below.

**The role of softmax**   The key property that leads to the above result is that the softmax operation in the attention mechanism cannot fundamentally disconnect any directed edge in the graph $\mathcal{G}$. As a result, center nodes (Definition 3.1), which appear in the context directly or indirectly for all tokens in the sequence, will eventually gain a dominant role in the context as their direct and indirect contributions propagate through the graph. Empirically, Gu et al. (2025); Xiao et al. (2024) found that changing softmax to ReLU, which can disconnect edges in the graph, indeed mitigates the emergence of attention sinks.

**How No PE induces positional information**   Previous works have observed that the causal mask alone amplifies the position bias (Wang et al., 2024; Yu et al., 2024). Despite these observations, it remains insufficiently understood how the causal mask captures positional information and what information is being captured. One hypothesis in Kazemnejad et al. (2023) suggests that the causal mask may be simulating either an absolute PE or a relative PE with specific weight matrices.

Theorem 4.1 offers a different perspective. The causal mask results in earlier tokens being utilized more frequently during computation, inducing a sequential order. This bias aligns with the token order in the sequence. To validate this perspective, we present additional experimental results in Section 5.2, providing empirical evidence that the causal mask is not simulating any PE but instead just exhibits a bias toward the earlier parts of a sequence.

**Trade-off between representation power and position bias**   Theorem 4.1 also highlights a trade-off between representational power and positional bias as the depth of attention layers increases. While numerous studies have demonstrated that deeper attention models are crucial for improving representation power (Li et al., 2024b; Merrill & Sabharwal, 2022; Sanford et al., 2024; Yun et al., 2020a), our findings reveal that these benefits come at a cost. As the model depth increases, the initial tokens in a sequence are utilized more frequently, amplifying the positional bias toward the beginning of the sequence. This trade-off underscores the importance of carefully balancing depth and positional bias in the design of attention-based architectures.

Theorem 4.1 on the causal mask can be generalized to encompass other types of attention masks, notably the sliding-window mask (Beltagy et al., 2020; Jiang et al., 2023) and the prefix mask (Lewis et al., 2020; Raffel et al., 2020). In the sliding-window mask, each token is allowed to attend to a fixed number of preceding tokens. Let $w$ denote the width of the sliding-window, representing the maximal number of tokens each token can access. The following result shows how limiting the context window size affects the propagation of contextual information in attention mechanism.

**Theorem 4.2.** *Let $\mathcal{G}$ be the sliding-window mask with width $w \geq 2$. Under* **A1**-**A2***, given $X^{(0)} \in \mathbb{R}^{N \times d}$, for every token $i \in [N]$,*

$$\lim_{t \to \infty} \mathbb{P}^{(t)}(z_i = 1 | X^{(0)}) = 1 \,.$$

*Moreover, there exist $0 < C, \epsilon < 1$ where $N\epsilon^{\left\lceil \frac{N-1}{w-1} \right\rceil} < 1$ such that*

$$\mathbb{P}^{(t)}(z_i = j | X^{(0)}) \leq C(1 - (j-1)\epsilon^{\left\lceil \frac{N-1}{w-1} \right\rceil})^{t/\left(2\left\lceil \frac{N-1}{w-1} \right\rceil\right)} \,.$$

*for all $1 < j \leq i$ and $t \geq 0$.*

The above result suggests that a smaller window size $w$ helps mitigate the model's bias toward early tokens in the sequence. However, such a moderating effect has its limit – the contextual information will still exponentially converge toward the first token over successive layers, though at a rate determined by the ratio between the sequence length $N$ and the window size $w$.

Finally, for the case of a prefix mask, where the first $K$ tokens in the sequence serve as a prefix and all subsequent tokens attend to them, contextual information exponentially converges toward these $K$ tokens rather than being dominated by just the first one, with each of these $K$ tokens having a non-trivial influence.

**Theorem 4.3.** *Let $\mathcal{G}$ be the prefix mask with the first $K$ tokens being the prefix tokens. Under* **A1**-**A2***, given $X^{(0)} \in \mathbb{R}^{N \times d}$, for every token $i \in [N]$,*

$$\lim_{t \to \infty} \mathbb{P}^{(t)}(z_i \in [K] | X^{(0)}) = 1 \,,$$

*and there exists $\kappa > 0$ such that*

$$\liminf_{t \to \infty} \mathbb{P}^{(t)}(z_i = k | X^{(0)}) \geq \kappa. \qquad \forall k \in [K] \,.$$

*Moreover, there exist $0 < C, \epsilon < 1$ where $N\epsilon < 1$ such that*

$$\mathbb{P}^{(t)}(z_i = j | X^{(0)}) \leq C(1 - (j - K)\epsilon)^t \,.$$

*for all $K < j \leq i$ and $t \geq 0$.*

**Attention sink and center node**   The above result connects the emergence of attention sinks to the structural role of center nodes in the graph $\mathcal{G}$ defined by the mask. Specifically, in Gu et al. (2025), the authors observed two interesting phenomena: 1) when using the sliding-window mask, attention sinks still appear on the absolute first token in the sequence, but not on the first token within each context window; 2) when using the prefix mask, attention sinks emerge on all prefix tokens, rather than just on the first token.

These empirical results align well with Theorems 4.2 and 4.3. Our results suggest that the absolute first token and the prefix tokens act as center nodes for the sliding-window mask and prefix mask, respectively. The context for each

token, after multi-layer attention, exponentially converges to these center nodes. This connection between attention sinks and center nodes suggests that attention sinks are not arbitrary artifacts but arise naturally from the underlying graph structure induced by the attention mask.

## 4.2. Relative PEs: A Competing Decay Effect

Having analyzed how attention masks bias the model toward the beginning of the sequence, we now shift our focus to studying PEs, the other key mechanism for representing positional information in transformers.

Relative PE, as the name suggests, incorporates positional information by modifying the original attention scores in a way that reflects the relative positions of tokens. Among these, the decay mask (Press et al., 2022) explicitly introduces a distance-based decay effect into the attention mechanism. We begin by examining the effect of the decay mask on individual attention layers.

**Lemma 4.4.** *Consider the decay mask in (2) where $\mathcal{G}$ is causal. Under **A1**-**A2**, given $X^{(0)} \in \mathbb{R}^{N \times d}$, there exists $C_{\max}, C_{\min} > 0$ such that for all $j \leq i \in [N]$ and $t \geq 0$,*

$$C_{\min} e^{-(i-j)m} \leq (A_{\text{decay}}^{(t)})_{ij} \leq C_{\max} e^{-(i-j)m}.$$

Lemma 4.4 demonstrates that the decay mask introduces an exponential decay effect into each attention map, with the strength of the effect determined by the token distances. However, while this result characterizes the behavior of individual attention layers, the interaction between layers in a multi-layer setting leads to more intricate behaviors. Building on Lemma 4.4 , Theorem 4.5 examines the cumulative effect of the decay mask across multiple layers when combined with the causal mask.

**Theorem 4.5.** *Consider the decay mask in (2) where $\mathcal{G}$ is causal. Fix $T \geq 0$. Under **A1**-**A2**, given $X^{(0)} \in \mathbb{R}^{N \times d}$, it holds for all $j \leq i \in [N]$ and $t \leq T$,*

$$\mathbb{P}_{\text{decay}}^{(t)}(z_i = j | X^{(0)}) = \Theta\left(\binom{t+i-j}{i-j} e^{-(i-j)m}\right).$$

Notably, if we denote

$$L(x) = \log\left(\binom{t+x}{x} e^{-xm}\right),$$

then $L(x)$ is not a monotone function of the distance $x$ between two tokens. More precisely, under Stirling's approximation, the critical point, where the highest attention score occurs, is at $x^* = t/(e^m - 1)$. This means that increasing the decay strength $m$ decreases $x^*$, making the model more biased towards recent tokens, whereas increasing the number of attention layers increases $x^*$, making the model more biased towards initial tokens.

Compared to Lemma 4.4, while the decay mask imposes a stronger decay effect on earlier tokens within individual attention layers, these tokens gain more cumulative importance across multiple layers. This trade-off between layer-wise decay and cross-layer accumulation transforms the initially monotonic decay pattern within each attention map into a more intricate, non-monotonic behavior when aggregated throughout the network.

## 4.3. A Closer Look at RoPE

Having analyzed the effect of the decay mask, which directly incorporates a distance-based decay into the attention score calculation, we now turn our attention to another popular form of relative positional encoding: RoPE (Su et al., 2023).

RoPE's inherent complexity has made a clear theoretical understanding challenging. However, recent empirical observations in Barbero et al. (2024b) suggest that in practice, LLMs tend to predominantly utilize feature dimensions that rotate slowly. This phenomenon introduces additional structure, enabling a more refined analysis of RoPE's effects by focusing on these slowly rotating feature dimensions. For simplicity and without loss of generality, we consider the case where only the slowest-rotating feature dimensions with base rotational angle $\theta_1$ are used by the model, i.e. effectively reducing the embedding dimension to $d = 2$. See Appendix I for more results on the general case $d \geq 2$.

Recall from (3) that RoPE operates by rotating the original query and key embeddings by an angle proportional to the token's position index within the sequence. Similar to the decay mask, which incorporates distance-based decay into attention scores, RoPE adjusts raw attention scores via these rotations. To formalize this relationship mathematically, we define the original angle between query $q_i^{(t)} := X_{i,:}^{(t)} W_Q^{(t)}$ and key $k_j^{(t)} := X_{j,:}^{(t)} W_K^{(t)}$ as $\phi_{i,j}^{(t)}$. Then the following result analyzes how RoPE's position-dependent rotations systematically modify the computation of attention scores.

**Lemma 4.6.** *Let $\mathcal{G}$ be the causal mask and $d = 2$. Suppose for $t \geq 0$, $\|q_i^{(t)}\|_2, \|k_j^{(t)}\|_2 > 0$, and $|\phi_{i,j}^{(t)}| \leq \delta\theta_1$, where $\delta > 0$ and $(\delta + N - 1)\theta_1 \leq \pi$. Then under **A1**-**A2**, given $X^{(0)} \in \mathbb{R}^{N \times d}$, there exists $C_{\max}, C_{\min}, c, c' > 0$ such that for all $j \leq i \in [N]$,*

$$C_{\min} e^{-c(i-j)^2 \theta_1^2} \leq (A_{\text{RoPE}}^{(t)})_{ij} \leq C_{\max} e^{-c'(i-j)^2 \theta_1^2}.$$

The result shows that by solely leveraging feature dimensions that rotate slowly, RoPE effectively induces a distance-based decay effect, which aligns with the intuition in Su et al. (2023). However, it is worth noting that the decay effect induced by RoPE is significantly smaller compared to that of the decay mask (Lemma 4.4). This is because the base rotational angle $\theta_1$ is typically chosen to be small, i.e. $\approx 1/10000$ per token (Dubey & et al., 2024; Su et al., 2023), resulting in a more gradual decay.

However, similar to the case of the decay mask, when considering the effect of RoPE across multiple layers of attention, the long-term decay effects within individual layers are counteracted by the increasing influence of earlier tokens given by the causal mask.

**Theorem 4.7.** *Fix $T > 0$. Under the same conditions as in Lemma 4.6 for $t \leq T$, given $X^{(0)} \in \mathbb{R}^{N \times d}$, there exists $c > 0$ such that for all $j \leq i \in [N]$ and $t \leq T$,*

$$\mathbb{P}^{(t)}_{\text{RoPE}}(z_i = j | X^{(0)}) = \Theta \left( \binom{t + i - j}{i - j} e^{-c(i-j)^2 \theta_1^2} \right).$$

Again, if we write

$$L(x) = \log \left( \binom{t + x}{x} e^{-x^2 \theta_1^2} \right),$$

then, by implicit differentiation, the critical point $x^*$ is an increasing function of the depth $t$ and a decreasing function of the base rotational angle $\theta_1$ (see Appendix H). This implies that increasing the base rotational angle $\theta_1$ reduces the optimal distance $x^*$, amplifying the long-term decay effect and causing tokens to focus more on nearby tokens. In contrast, increasing the number of attention layers $t$ increases $x^*$ and hence deeper models become more biased toward initial tokens.

## 5. Experiments

In this section, we validate our theoretical findings via carefully designed numerical experiments[1]. To ensure a controlled setup that enables precise manipulation of positional biases in the data, we adopt the synthetic data-generating process and simplified self-attention network framework proposed in Reddy (2024). This setup allows us to systematically isolate and examine the effects of different components on the emergence of position bias.

**Task structure**   Following Reddy (2024), we adopt the following information retrieval task: The model is trained to predict the label $y_{\text{query}}$ of a target $x_{\text{query}}$ using the cross-entropy loss, given an alternating sequence of $n$ items and $n$ labels: $x_1, y_1, \ldots, x_n, y_n, x_{\text{query}}$. The sequence is embedded in $d$ dimensions. Each $x_i$ is sampled from a Gaussian mixture model with $K$ classes (see Appendix L for results under a fixed-vocabulary setup), and $y_i$ is the corresponding class label assigned prior to training from the total $L$ labels ($L \leq K$). The burstiness $B$ is the number of occurrences of $x_i$ from a particular class in an input sequence. Importantly, at least one item in the context belongs to the same class as the query. To control position bias in the training data, $x_{\text{query}}$ can either be explicitly assigned to the class of a

---

[1]Our code is available at github.com/xinyiwu98/position-bias-in-attention.

specific $x_i$, introducing position-dependent bias in the data, or randomly assigned to the class of any $x_i$, simulating a scenario without position bias in the data.

**Tracking position bias**   To quantify position bias, we evaluate model performance using sequences containing novel classes not seen during training. Specifically, by generating new class centers for the Gaussian mixture and randomly assigning one of the $L$ existing labels to these novel classes, we ensure that the model relies on contextual information rather than memorized class features. Crucially, we can systematically vary the position of the correct answer within test sequences to measure retrieval accuracy changes, thereby isolating and quantifying position-dependent biases in the model's behavior.

**Network architecture**   The input sequences are passed through an attention-only network followed by a classifier. Each attention layer has one attention head. The classifier is then a three-layer MLP with ReLU activations and a softmax layer which predicts the probabilities of the $L$ labels.

Following Reddy (2024), we set $n = 8$ and $d = 64$. Additional experimental details are provided in Appendix J. Despite our use of a simplified experimental setup, we observe the emergence of key phenomena documented in real-world LLMs, such as the "lost-in-the-middle" phenomenon (Section 5.2) and the formation of attention sinks (Appendix K.4). This convergence between our controlled environment and real-world observations validates our choice of abstraction, suggesting that we have preserved the essential mechanisms driving position bias while enabling systematic investigation.

### 5.1. The Effects of Depth and Relative PEs

To investigate the position bias arising solely from the architectural design of the attention mechanism, we use training sequences without positional bias, where the position of $x_i$ sharing the same class as $x_{\text{query}}$ is uniformly random in $\{1, 2, \ldots, n\}$. We evaluate two architectural variants: one without residual connections, and one with standard residual connections applied after each attention layer. The no-residual variant serves as a theoretical probe, isolating the contribution of attention alone to positional bias. In contrast, the residual variant reflects standard architectural design and enables us to examine how residual connections, in addition to attention, influence the emergence and amplification of positional bias.

To evaluate the position bias in the trained model, we construct test sequences of the form $[\boldsymbol{a}, b]$. Here, the bolded term $\boldsymbol{a}$ explicitly marks the correct position, ensuring $y_a$ matches $y_{\text{query}}$, while position $b$ serves as a baseline. In these sequences, $x_a$ and $x_b$ are identical vectors, allowing us to control for the influence of semantic information on the

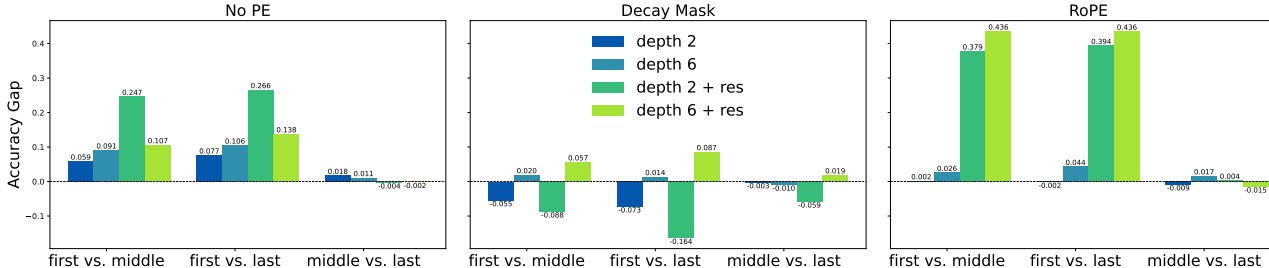

Figure 2: Position bias arising solely from the architectural design of the attention mechanism, with **no positional bias in the training data**. $a$ vs. $b$ denotes the gap for the case $[\boldsymbol{a}, b] - [\boldsymbol{b}, a]$, where bar magnitude indicates gap size, positive indicates bias toward earlier position, and negative indicates bias toward later position. Deeper attention amplifies the bias toward earlier tokens, regardless of the PE used. Furthermore, decay mask introduce stronger distance-based decay effects that increase focus on recent tokens than RoPE. Residual connections affect positional bias in a nontrivial, depth- and PE-dependent way (see Appendix K.2 for more details).

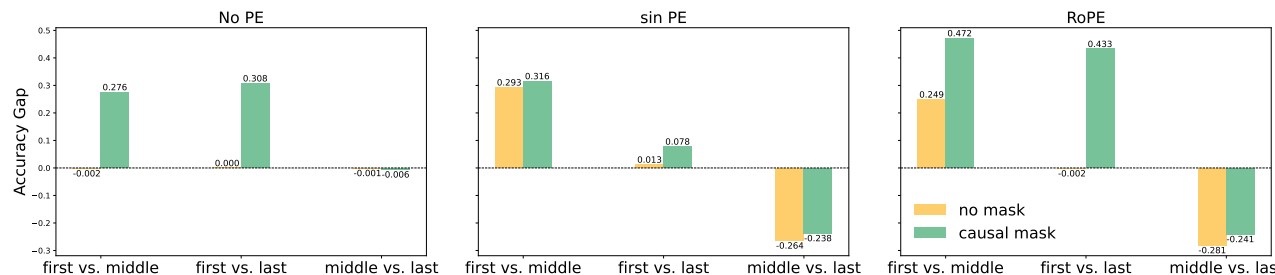

Figure 3: Position bias when **trained on data biased toward the first and last positions**. Compared with no mask, a causal mask without PE indeed introduces positional dependencies. However, pure causal mask captures positional bias only at the first position but not at the last, whereas both sin PE and RoPE successfully capture biases at both ends. Moreover, the performance under PEs also displays a "lost-in-the-middle" pattern, which is absent under other types of positional bias in the training data (see Appendix K.3 for more details).

model's retrieval accuracy. We then measure the retrieval accuracy gap between pairs of sequences where the content at positions $a$ and $b$ is identical, but the correct position varies. This gap, defined as $[\boldsymbol{a}, b] - [\boldsymbol{b}, a]$, quantifies the model's positional preference independent of semantic information. To perform this evaluation, we construct three pairs of test sets, each containing $10,000$ sequences: [**first**, middle] vs. [**middle**, first], [**first**, last] vs. [**last**, first], and [**middle**, last] vs. [**last**, middle]. Here "first" (position 1), "middle" (position $n/2$), and "last" (position $n$) denote fixed positions within a sequence.

Figure 2 shows the average results over five runs, where $a$ vs. $b$ denotes the gap $[\boldsymbol{a}, b] - [\boldsymbol{b}, a]$. The magnitude of each bar represents the size of the performance gap, and the sign of each bar reflects the direction of the bias: a positive sign indicates a bias toward earlier positions, while a negative sign indicates a bias toward later positions. We highlight several key observations. First, without residual connections, increasing model depth consistently amplifies the bias toward earlier parts of the sequence, regardless of the PE used (see Appendix K.1 for additional results). Also note that the performance gap between the middle and last positions is notably smaller than that involving the first position. This aligns with our theory, which suggests that as the model focuses more on the initial part of the

sequence, information near the sequence's end becomes less distinguishable, consistent with the patterns observed in Barbero et al. (2024a). Furthermore, both the decay mask and RoPE introduce distance-based decay effects that reduce the bias toward the beginning induced by the causal mask and increase the focus on recent tokens. However, the decay effect induced by the decay mask is substantially more pronounced than that by RoPE, as predicted by our theory.

**Effect of residual connections on position bias** Residual connections appear to interact with positional bias in a nontrivial way. While position bias still emerges in models with residual connections, its relationship with depth becomes non-monotonic and can be either amplified or reduced, depending on the positional encoding used and the depth regime. We provide a more detailed analysis in Appendix K.2.

## 5.2. Can Causal Mask Induce Usable Positional Information?

Next, we empirically examine how the causal mask leverages positional information. Kazemnejad et al. (2023) hypothesized that without PE (No PE), the causal mask can implicitly simulate absolute or relative PE through specific

weight matrix configurations. To test this hypothesis, we train models with residual connections on sequences with positional bias at either the beginning or the end. Specifically, in the training data, $x_{\text{query}}$ is assigned to the class of $x_1$ or $x_n$ with equal probability. We then evaluate two types of attention masks: no mask ($\mathcal{G}$ is complete) and causal, and three types of PEs: No PE (Kazemnejad et al., 2023), absolute sinusoidal PE (sin PE) (Vaswani et al., 2017), and relative PE using RoPE. For evaluation, we construct six types of test sets as described in Section 5.1, each with $10,000$ sequences.

Figure 3 shows the average results using a 2-layer network over five runs. Notably, in the left subplot, the causal mask without PE demonstrates a clear position bias toward the first position compared to the no mask without PE. This indicates that the causal mask indeed introduces a notion of position. However, when strong positional biases are present in the training data, both sin PE and RoPE allow the model to effectively capture these biases at both ends, regardless of the mask used. In contrast, a causal mask without PE only enables the model to learn a position bias at the beginning of the sequence. If the hypothesis by Kazemnejad et al. (2023) were correct, that the causal mask uses positional information by simulating PEs, then the model should be able to capture positional bias at any location. This discrepancy suggests that the causal mask does not inherently implement PE but instead introduces a bias toward earlier positions via iterative attention, capturing positional bias only when it aligns with this predisposition.

**The role of data in creating positional bias**    It is worth noting that in Figure 3, we observe the "lost-in-the-middle" phenomenon (Liu et al., 2024), where information retrieval accuracy follows a U-shape relative to the position of the answer, with performance at the beginning of the sequence slightly better than at the end. More experimental results under different types of positional bias in the training data can be found in Appendix K.3. Notably, this phenomenon does not occur when the training data lacks positional bias (Figure 2)[2] or contains other types of positional bias considered (Appendix K.3). This suggests that specific types of positional bias in the training data also play a role in how the model learns to prioritize positions within a sequence.

An additional observation from the fixed-vocabulary setting (see Appendix L for details) is that the geometry of the token embeddings can significantly influence task difficulty. This geometric structure may also affect the emergence and nature of positional bias, as it shapes how the model learns to distinguish and attend to tokens.

---

[2]With the sole exception occurring in the deep-layer regime with RoPE; see Appendix K.2.

# 6. Conclusion

In this paper, we study position bias in transformers through a probabilistic and graph-theoretic lens, developing a theoretical framework that quantifies how positional information influences context construction across multi-layer attention. Our analysis reveals two key findings about position bias in transformers: the causal mask's inherent bias toward earlier tokens, as deeper layers increasingly attend to these positions through iterative attention, and the interplay between causal masking and relative positional encodings, which results in a nuanced, non-monotonic balance between distance-based decay effects and the cumulative influence of earlier positions. These findings open several promising directions for future work. One potential direction is leveraging these insights to design bias-free transformers, mitigating positional biases to improve model robustness and generalization capabilities. Alternatively, our framework can also inform the strategic exploitation of positional bias in specific applications, such as emphasizing early positions for text summarization or prioritizing recent interactions in recommendation systems. Another important direction is to extend our analysis to other transformer components— such as residual connections, value projections, and MLPs— which may interact with attention in nontrivial ways, as suggested by our results in Section 5.1 and analyses based on vector norms (Kobayashi et al., 2020). By deepening our understanding of how architectural choices in transformers shape positional dependencies, our work provides a foundation for designing attention mechanisms with predictable and task-aligned positional properties.

# Acknowledgements

The authors would like to thank Gautam Reddy for the help with the experimental setup and acknowledge the MIT SuperCloud and Lincoln Laboratory Supercomputing Center for providing computing resources that have contributed to the research results reported within this paper.

XW and AJ were supported by ONR N00014-23-1-2299. YW and SJ were supported by ONR N00014-20-1-2023 (MURI ML-SCOPE), NSF CCF-2112665 (TILOS AI Institute), and an Alexander von Humboldt Professorship.

# Impact Statement

This paper presents work whose goal is to advance the field of Machine Learning. There are potential societal consequences of our work, none which we feel must be specifically highlighted here.

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

# A. Proof of Theorem 4.1

## A.1. Auxiliary results

**Lemma A.1.** *Under* **A1**-**A2**, *there exists* $\epsilon > 0$ *such that* $A_{ij}^{(t)} \geq \epsilon$ *for all* $t \geq 0$, $(j, i) \in E$.

*Proof.* Writing (1) recursively, we get that the token trajectories

$$X^{(t+1)} = A^{(t)}...A^{(0)}X^{(0)}W_V^{(0)}...W_V^{(t)}, \tag{5}$$

stay uniformly bounded for all $t \geq 0$ by **A2**. Then it follows from **A1** that there exists $C \in \mathbb{R}$ such that for all $t \geq 0$,

$$\left\| \left( X^{(t)}W_Q^{(t)} \right)_{i,:} \right\|_2 = \left\| X_{i,:}^{(t)}W_Q^{(t)} \right\|_2 \leq C,$$
$$\left\| \left( X^{(t)}W_K^{(t)} \right)_{i,:} \right\|_2 = \left\| X_{i,:}^{(t)}W_K^{(t)} \right\|_2 \leq C. \tag{6}$$

Hence for all $i, j \in [N]$,

$$-C^2 \leq (X^{(t)}W_Q^{(t)}(X^{(t)}W_K^{(t)})^\top)_{ij} \leq C^2.$$

This implies that there exists $\epsilon > 0$ such that $A_{ij}^{(t)} \geq \epsilon$ for all $(j, i) \in E$. $\qquad\square$

## A.2. Proof of Theorem 4.1

We denote $P^{(t)} := A^{(t)} \cdots A^{(0)}$. It suffices to show that there exists $0 < C < 1$ and $0 < \epsilon < 1$ such that

$$P_{ij}^{(t)} \leq C(1 - (j-1)\epsilon)^t \tag{7}$$

for all $1 < j \leq i$ and $t \geq 0$.

The proof will go by induction:

**Base case**  By Lemma A.1, it follows that

$$P_{ij}^{(0)} \leq (1 - \epsilon)$$

for all $1 < j \leq i$. Then let $C := 1 - \epsilon$.

**Induction step**  Assume that (7) holds, it follows that for all $1 < j \leq i$.

$$P_{ij}^{(t+1)} = \sum_{k=j}^{i} A_{ik}^{(t)} P_{kj}^{(t)} \leq (1 - (j-1)\epsilon)C(1 - (j-1)\epsilon)^t = C(1 - (j-1)\epsilon)^{t+1}.$$

From above, we conclude the theorem.

# B. Proof of Theorem 4.2

For $t_0 \leq t_1$, we denote

$$A^{(t_1:t_0)} = A^{(t_1)} \cdots A^{(t_0)}.$$

Without loss of generality, we assume in the following proof that $N - 1$ can be divided by $w - 1$.

## B.1. Auxiliary results

**Lemma B.1.** *Let* $\mathcal{G}$ *be the sliding-window mask with the window size* $w \geq 2$. *Then there exists* $c > 0$ *such that for all* $t_0 \geq 0$,

$$c \leq A_{ij}^{\left(t_0 + \frac{N-1}{w-1} - 1:t_0\right)} \leq 1, \quad \forall j \leq i \in [N].$$

*Proof.* Given the connectivity pattern of the sliding-window mask $\mathcal{G}$ and Lemma A.1, it follows that for all $t_0 \geq 0$, $A^{\left(t_0 + \frac{N}{w} - 1 : t_0\right)}$ is a lower triangular matrix. Moreover, since $A_{ij}^{\left(t_0 + \frac{N}{w} - 1 : t_0\right)}$ counts the aggregate probability of the walks of length $\frac{N-1}{w-1}$ between token $i$ and token $j$ where by Lemma A.1, each walk has probability at least $\epsilon^{\frac{N-1}{w-1}}$.

Thus we conclude that there exists $c > 0$ such that for all $t_0 \geq 0$,

$$A_{ij}^{\left(t_0 + \frac{N}{w} - 1 : t_0\right)} \geq c, \quad \forall j \leq i \in [N].$$

$\square$

## B.2. Proof of Theorem 4.2

For $k \geq 0$, denote

$$\tilde{A}^{(k)} = A^{\left((k+1)\left(\frac{N-1}{w-1}\right) - 1 : k\left(\frac{N-1}{w-1}\right)\right)}$$

and

$$\tilde{P}^{(k)} = \tilde{A}^{(k)} \cdots \tilde{A}^{(0)} .$$

Then by Lemma B.1 and Theorem 4.1, we get that there exists $0 < C < 1$ and $0 < c < 1$ such that for all $k \geq 0$

$$\tilde{P}_{ij}^{(k)} \leq C(1 - (j-1)c)^k , \quad \forall j \leq i \in [N] .$$

Denote $Q_j^{(t)} = \max_{1 \leq i \leq N} P_{ij}^{(t)}$ and $\tilde{Q}_j^{(k)} = \max_{1 \leq i \leq N} \tilde{P}_{ij}^{(k)}$. Then it follows that for all $k \geq 0$,

$$\tilde{Q}_j^{(k)} \leq C(1 - (j-1)c)^k.$$

Observe that

$$\forall i \in [N], P_{ij}^{(t)} \leq Q_j^{(t)} , \tag{8}$$

and

$$\forall j \in [N], Q_j^{(t+1)} \leq Q_j^{(t)} . \tag{9}$$

Let $q_j = (1 - (j-1)c)^{2\frac{1}{\frac{N-1}{w-1}}}$. Then for all $k \geq 1$ and $0 \leq r < \frac{N-1}{w-1}$,

$$q_j^{k\left(\frac{N-1}{w-1}\right) + r} \geq (1 - (j-1)c)^k.$$

This implies that for all $t \geq \frac{N-1}{w-1}$,

$$P_{ij}^{(t)} \leq q_j^t = C(1 - (j-1)c)^{t/\left(2\frac{N-1}{w-1}\right)} .$$

As for $t < \frac{N-1}{w-1}$, notice that $P_{ij}^{(0)} \leq 1 - \epsilon$ for all $j \leq i \in [N]$ by Lemma A.1. Then by 8 and 9, we deduce that

$$P_{ij}^{(t)} \leq (1 - \epsilon)^{\frac{t+1}{\frac{N-1}{w-1}}} , \quad \forall j \leq i \in [N].$$

We thus conclude the statement.

# C. Proof of Theorem 4.3

First note that the first and third statements:

$$\lim_{t\to\infty} \mathbb{P}^{(t)}(z_i \in [K]|X^{(0)}) = 1,\tag{10}$$

for all $\in [N]$ and

$$\mathbb{P}^{(t)}(z_i = j|X^{(0)}) \leq C(1 - (j - K)\epsilon)^t.$$

for all $K < j \leq i$ and $t \geq 0$, follow immediately from Theorem 4.1 by regarding the first K tokens as a super node in the causal graph $\mathcal{G}$ and aggregate the edges accordingly. Thus it suffices to show that there exists $\kappa > 0$ such that

$$\liminf_{t\to\infty} \mathbb{P}^{(t)}(z_i = k|X^{(0)}) \geq \kappa. \qquad \forall k \in [K].\tag{11}$$

For $t > 0$, consider

$$P_{ik}^{(t)} = \sum_{l=1}^{N} P_{il}^{(t:1)} A_{lk}^{(0)}.$$

Then for $k_1, k_2 \in [K]$,

$$\frac{P_{ik_1}^{(t)}}{P_{ik_2}^{(t)}} = \frac{\sum_{l=1}^{\max\{i,K\}} P_{il}^{(t:1)} A_{lk_1}^{(0)}}{\sum_{l=1}^{\max\{i,K\}} P_{il}^{(t:1)} A_{lk_2}^{(0)}},$$

which then follows

$$\frac{P_{ik_1}^{(t)}}{P_{ik_2}^{(t)}} \geq \min_{1 \leq l \leq \min\{i,K\}} \frac{A_{lk_1}^{(0)}}{A_{lk_2}^{(0)}}.$$

Then by Lemma A.1, there exists $C > 0$ such that for all $k_1, k_2 \in [K]$,

$$C \leq \liminf_{t\to\infty} \frac{P_{ik_1}^{(t)}}{P_{ik_2}^{(t)}}.$$

Since $\lim_{t\to\infty} \sum_{k=1}^{K} P_{ik}^{(t)} = 1$ by (10), we deduce (11) as desired.

# D. Proof of Lemma 4.4

Fix $t \geq 0$. Let

$$Z_{ij}^{(t)} = (X^{(t)}W_Q^{(t)})_{i,:}(X^{(t)}W_K^{(t)})_{:,j}.$$

Following from Lemma A.1, there exists $I_{\min}, I_{\max} \in \mathbb{R}$ such that for all $j \leq i \in [N]$,

$$Z_{ij}^{(t)} \in [I_{\min}, I_{\max}].$$

Consider the denominator in the $\mathrm{softmax}(\cdot)$ operation in the calculation of $(A_{\mathrm{decay}}^{(t)})_{ij}$:

$$\sum_{k=1}^{i} e^{Z_{ik}^{(t)} - (i-k)m} \geq e^{I_{\min}} \sum_{k=0}^{i} e^{-(i-k)m}$$
$$= e^{I_{\min}} \frac{1 - e^{-(i+1)m}}{1 - e^{-m}}$$
$$\geq e^{I_{\min}} \frac{1 - e^{-2m}}{1 - e^{-m}}$$
$$= e^{I_{\min}}(1 + e^{-m})$$

and

$$\sum_{k=1}^{i} e^{Z_{ik}^{(t)} - (i-k)m} \leq e^{I_{\max}} \sum_{k=0}^{\infty} e^{-km}$$

$$= \frac{e^{I_{\max}}}{1 - e^{-m}}$$

It follows that

$$(A_{\text{decay}}^{(t)})_{ij} \leq \frac{e^{I_{\max} - (i-j)m}}{e^{I_{\min}}(1 + e^{-m})} = C_{\max} e^{-(i-j)m}$$

and

$$(A_{\text{decay}}^{(t)})_{ij} \geq \frac{e^{I_{\min} - (i-j)m}}{e^{I_{\max}}/(1 - e^{-m})} = C_{\min} e^{-(i-j)m}$$

where $C_{\max} := e^{(I_{\max} - I_{\min})}/(1 + e^{-m})$ and $C_{\min} := (1 - e^{-m})e^{(I_{\min} - I_{\max})}$.

## E. Proof of Theorem 4.5

Note that in the causal graph $\mathcal{G}$, there are $\binom{t+i-j}{i-j}$ paths of length $t+1$ from token $j$ to token $i$.

Since going from token $j$ to token $i$ in the causal graph, the connectivity patterns ensure that the token indices along the path are non-decreasing, i.e. if we denote the directed path as $(j, l_1), (l_1, l_2), ..., (l_t, i)$, it holds that $j \leq l_1 \leq l_2 \leq ... \leq l_t \leq i$. Together with Lemma 4.4, we conclude the theorem statement.

## F. Proof of Lemma 4.6

Fix $t \geq 0$. Denote the angle after rotation to be $\psi_{i,j}^{(t)}$. Then it follows from the definition of RoPE that

$$\psi_{i,j}^{(t)} = \phi_{i,j}^{(t)} - (i-j)\theta_1 \,.$$

Thus

$$|\psi_{i,j}^{(t)}| = |\phi_{i,j}^{(t)} - (i-j)\theta_1| \geq ||(i-j)\theta_1| - |\phi_{i,j}^{(t)}|| \geq |(i-j) - \delta|\theta_1 \,.$$

$$|\psi_{i,j}^{(t)}| = |\phi_{i,j}^{(t)} - (i-j)\theta_1| \leq |(i-j)\theta_1| + |\phi_{i,j}^{(t)}| \leq (i-j+\delta)\theta_1 \,.$$

Let the original query $i$ and key $j$ embeddings be $q_i^{(t)} := X_{i,:}^{(t)} W_Q^{(t)}$ and $k_j^{(t)} := X_{j,:}^{(t)} W_K^{(t)}$, respectively, and the corresponding query $i$ and key $j$ embeddings after rotation be $q_i'^{(t)}$ and $k_j'^{(t)}$, respectively.

Since $\langle q_i'^{(t)}, k_j'^{(t)} \rangle = \|q_i^{(t)}\|_2 \|k_j^{(t)}\|_2 \cos \psi_{i,j}^{(t)}$, it follows from that there exists $C_{\min}, C_{\max} \geq 0$ such that for all $i, j \in [N]$,

$$C_{\min} \cos((i - j + \delta)\theta_1) \leq \langle q_i'^{(t)}, k_j'^{(t)} \rangle \leq C_{\max} \cos(|(i-j) - \delta|\theta_1) \,.$$

Since for $|x| \leq \pi$ there exists $c > 0$ such that $1 - x^2/2 \leq \cos x \leq 1 - x^2/c$, we get that

$$C_{\min} \left( 1 - \frac{((i-j) + \delta)^2 \theta_1^2}{2} \right) \leq \langle q_i'^{(t)}, k_j'^{(t)} \rangle \leq C_{\max} \left( 1 - \frac{((i-j) - \delta)^2 \theta_1^2}{c} \right) \,.$$

and hence

$$C_{\min} \left( 1 - \delta^2 \theta_1^2 - (i-j)^2 \theta_1^2 \right) \leq \langle q_i'^{(t)}, k_j'^{(t)} \rangle \leq C_{\max} \left( 1 - \frac{((i-j)^2/2 - \delta^2)\theta_1^2}{c} \right) \,.$$

Consider $Y_i^{(t)} = \sum_{k=1}^{i} e^{\langle q_i'^{(t)}, k_j'^{(t)} \rangle}$. Then by (6), we get that there exists $Y_{\max}, Y_{\min} > 0$ such that

$$Y_{\max} \leq Y_i^{(t)} \leq Y_{\min} \,.$$

We thus conclude the statement.

# G. Proof of Theorem 4.7

Notice that

$$(P_{\text{RoPE}}^{(t)})_{ij} = \sum_{l_1 \le \cdots \le l_{t-1} \in [N]^{t-1}} A_{il_{t-1}}^{(t-1)} A_{l_{t-1}l_{t-2}}^{(t-2)} \cdots A_{l_1 j}^{(0)} \tag{12}$$

Given that when $\mathcal{G}$ is the causal graph, due to the connectivity the directed path of length $t$ from token $j$ to token $i$ must be non-decreasing, i.e. $j \le l_1 \le l_2 \le \cdots \le l_{t-1} \le i$, and there would be in total $\binom{t+i-j}{i-j}$ such paths. For each such path $j \le l_1 \le l_2 \le \cdots \le l_{t-1} \le i$, notice that by Lemma 4.6, we get that fix $T \ge 0$, there exists $C_{\min}, C_{\max} > 0$ such that

$$A_{i,l_{t-1}}^{(t-1)} A_{l_{t-1},l_{t-2}}^{(t-2)} \cdots A_{l_1,j}^{(0)} \ge C_{\min} e^{-c((i-l_{t-1})^2+(l_{t-1}-l_{t-2})^2+\cdots+(l_1-j)^2)\theta_1^2} \tag{13}$$

and

$$A_{i,l_{t-1}}^{(t-1)} A_{l_{t-1},l_{t-2}}^{(t-2)} \cdots A_{l_1,j}^{(0)} \le C_{\max} e^{-c'((i-l_{t-1})^2+(l_{t-1}-l_{t-2})^2+\cdots+(l_1-j)^2)\theta_1^2} \tag{14}$$

From (13), since $j \le l_1 \le l_2 \le \cdots \le l_{t-1} \le i$, we further get that

$$A_{i,l_{t-1}}^{(t-1)} A_{l_{t-1},l_{t-2}}^{(t-2)} \cdots A_{l_1,j}^{(0)} \ge C_{\min} e^{-c(i-j)^2\theta_1^2}, \tag{15}$$

and similarly

$$A_{i,l_{t-1}}^{(t-1)} A_{l_{t-1},l_{t-2}}^{(t-2)} \cdots A_{l_1,j}^{(0)} \le C_{\max} e^{-\frac{c'}{2}(i-j)^2\theta_1^2}. \tag{16}$$

# H. Implicit differentiation of $x$ with respect to $\theta_1$ and $t$

Recall that under RoPE,

$$L(x) = \log\left(\binom{t+x}{x} e^{-x^2\theta_1^2}\right).$$

Then by Stirling's approximation,

$$L(x) \approx ((t+x)\log(t+x) - (t+x)) - (x\log x - x) - \theta_1^2 x^2,$$

and thus

$$L'(x) = \log\left(\frac{t+x}{x}\right) - 2\theta_1 x.$$

Taking implicit differentiation of $t$:

$$\frac{\partial}{\partial t}\log\left(\frac{t+x}{x}\right) = \frac{-t}{x(x+t)}\frac{\partial x}{\partial t} + \frac{1}{x+t}$$

and

$$\frac{\partial}{\partial t} 2\theta_1 x = 2\theta_1 \frac{\partial x}{\partial t}.$$

So let

$$\frac{1}{x+t} = (2\theta_1 + \frac{t}{(x+t)x})\frac{\partial x}{\partial t}$$

and thus

$$\frac{\partial x}{\partial t} = \frac{1}{2\theta_1(x+t) + \frac{t}{x}} > 0.$$

Taking implicit differentiation of $\theta_1$:

$$\frac{\partial}{\partial\theta_1}\log\left(\frac{t+x}{x}\right) = \frac{-t}{x(x+t)}\frac{\partial x}{\partial\theta_1}$$

and

$$\frac{\partial}{\partial \theta_1} 2\theta_1 x = 2x + 2\frac{\partial x}{\partial \theta_1}\theta_1.$$

So let

$$\left(2\theta_1 + \frac{t}{x(x+t)}\right)\frac{\partial x}{\partial \theta_1} = -2x,$$

and thus

$$\frac{\partial x}{\partial \theta_1} = \frac{-2x}{2\theta_1 + \frac{t}{x(x+t)}} < 0.$$

Hence we observe that $x^*$ is an increasing function of $t$ and a decreasing function of $\theta_1$. This implies that increasing the base rotational angle $\theta_1$ reduces the optimal distance $x^*$, amplifying the long-term decay effect and causing tokens to focus more on nearby tokens. In contrast, increasing the number of attention layers $t$ increases $x^*$ and hence deeper models become more biased toward initial tokens.

# I. The effect of RoPE: case for $d \geq 2$

In this section, we present a generalized version of Theorem 4.7 for the case $d \geq 2$.

Let the query $q$ and key $k$ be vectors in $\mathbb{R}^d$, where $d$ is even, and let $\phi$ be the angle between $q$ and $k$, which we assume to be well-defined, with:

$$\cos \phi = \frac{\langle q, k \rangle}{\|q\|_2 \|k\|_2}.$$

Define the length-2 segments of query $q$ and and key $k$ as

$$q_l = (q_{2l-1}, q_{2l}), \quad k_l = (q_{2l-1}, q_{2l}),$$

for $l \in [d/2]$, and let $\phi_l$ be the angle between $q_l$ and $k_l$, with:

$$\cos \phi_l = \frac{\langle q_l, k_l \rangle}{\|q_l\|_2 \|k_l\|_2}.$$

Without loss of generality, we make the following assumption:

**A3** There exists $\beta_q, \beta_k > 0$ such that $\|q_l^{(t)}\|_2 \geq \beta_q \|q^{(t)}\|_2$ and $\|k_l^{(t)}\|_2 \geq \beta_k \|k^{(t)}\|_2$ for all $l \in [d/2]$ for all $t \geq 0$.

The condition means that all segments makes a nontrivial contribution to the norm. In practice, since LLMs tend to tend to predominantly utilize feature dimensions that rotate slowly (Barbero et al., 2024b), the effective $d/2$ tends to be a small number.

Given the pre-defined set of base rotational angles $\Theta = \{0 \leq \theta_1 \leq \cdots \leq \theta_{d/2}\}$, we reparametrize as $\theta_i = \alpha_i \theta_1$.

## I.1. Results

We present the general version of Lemma 4.6 and Theorem 4.7 as follows:

**Lemma I.1.** *Let $\mathcal{G}$ be the causal mask and* **A1-A3** *hold. Suppose for $t \geq 0$, $\|q_i\|_2, \|k_j\|_2 > 0$, and $|\phi_{i,j}^{(t)}| \leq \delta\theta_1$, where $\delta > 0$ and*

$$\left(\sqrt{\frac{1}{\beta_q \beta_k}}\delta\pi + 2(N-1)\alpha_{d/2}\right)\theta \leq 2\pi.$$

*Then there exists $C_{\max}, C_{\min}, c, c' > 0$ such that*

$$C_{\min} e^{-c\sum_{l=1}^{d/2}(i-j)^2\alpha_l^2\theta_1^2} \leq (A_{\text{RoPE}}^{(t)})_{i,j} \leq C_{\max} e^{-c'\sum_{l=1}^{d/2}(i-j)^2\alpha_l^2\theta_1^2}.$$

**Theorem I.2.** *Fix $T > 0$. Under the same conditions as in Lemma I.1 for $t \leq T$, there exists $c > 0$ such that for all $t \leq T$,*

$$(P_{\text{RoPE}}^{(t)})_{i,j} = \Theta\left( \binom{t+i-j}{i-j} e^{-c\sum_{l=1}^{d/2}(i-j)^2\alpha_l^2\theta_1^2} \right).$$

## I.2. Proofs of Lemma I.1

We first show the following auxiliary result:

**Lemma I.3.** *Under **A3**, it holds that*

$$|\phi_l| \leq \frac{\pi}{2}\sqrt{\frac{1}{\beta_q\beta_k}}|\phi|,$$

*for all $l \in [d/2]$.*

*Proof.* By definition, since

$$\sum_{l=1}^{d/2} \|q_l\|_2 \|k_l\|_2 \cos\phi_l = \|q\|_2 \|k\|_2 \cos\phi,$$

then the Cauchy–Schwarz inequality implies that

$$\sum_{l=1}^{d/2} \|q_l\|_2 \|k_l\|_2 (1 - \cos\phi_l) \leq \|q\|_2 \|k\|_2 (1 - \cos\phi). \tag{17}$$

By **A3**, (17) becomes

$$\sum_{l=1}^{d/2}(1 - \cos\phi_l) \leq \frac{1}{\beta_q\beta_k}(1 - \cos\phi).$$

Given the trigonometric identity $1 - \cos 2x = 2\sin^2 x$, we get that

$$\sum_{l=1}^{d/2} \sin^2\left(\frac{\phi_l}{2}\right) \leq \frac{1}{\beta_q\beta_k}\sin^2\left(\frac{\phi}{2}\right). \tag{18}$$

Notice for all $x \in \mathbb{R}$,

$$\sin^2 x \leq x^2, \tag{19}$$

and for all $|x| \leq \pi/2$,

$$\frac{4}{\pi^2}x^2 \leq \sin^2 x. \tag{20}$$

Apply (19) and (20) to (18), we get that

$$\sum_{i=1}^{d/2} \phi_l^2 \leq \frac{\pi^2}{4\beta_q\beta_k}\phi^2.$$

Hence for all $l \in [d/2]$, it follows that

$$|\phi_l| \leq \frac{\pi}{2}\sqrt{\frac{1}{\beta_q\beta_k}}|\phi|.$$

$\square$

Denote the angle after rotation to be $\psi_{i,j,l}$. Then it follows that

$$\psi_{i,j,l} = \phi_{i,j,l} - (i-j)\alpha_l\theta_1.$$

It follows similarly as in the proof of Lemma 4.6 that

$$C_{\min}\left(1 - \delta'^2\theta^2 - (i-j)^2\alpha_l^2\theta_1^2\right) \leq \langle (q_i')_l, (k_j')_l \rangle \leq C_{\max}\left(1 - \frac{((i-j)^2\alpha_l^2/2 - \delta'^2)\theta_1^2}{c}\right),$$

where $\delta' = \frac{\pi}{2}\sqrt{\frac{1}{\beta_q\beta_k}}\delta$, for all $l \in [d/2]$.

Since

$$\langle q_i', k_j' \rangle = \sum_{l=1}^{d/2} \langle (q_i')_l, (k_j')_l \rangle,$$

we conclude the statement.

### I.3. Proof of Theorem I.2

The result is a direct corollary of Lemma 4.6 and Theorem 4.7.

## J. Experiments

Here we provide more details on the numerical experiments presented in Section 5. All models were implemented with PyTorch (Paszke et al., 2019).

**Parameterizing the data distribution**    As defined in Section 5, the input data distribution is modulated by tuning various parameters. In addition to the parameters described in the main text, for the Gaussian mixture with $K$ classes, each class $k$ is defined by a $d$-dimensional vector $\mu_k$ whose components are sampled *i.i.d.* from a normal distribution with mean zero and variance $1/d$. Then the value of $x_i$ is given by $\mu_k + \epsilon\eta_i/\sqrt{1+\epsilon^2}$, where $\eta_i$ is drawn from the same distribution as the $\mu_k$'s and $\epsilon$ sets the within-class variability. Each class is assigned to one of $L$ labels ($L \leq K$). The contents of the labels are drawn prior to training from the same distribution as the $\mu_k$'s.

In Reddy (2024), the author found that different configurations of the data generating process give rise to different learning regimes. To enable better information retrieval ability of the model, we choose the configuration suggested by Reddy (2024) that corresponds to the difficult in-weight learning and easy in-context-learning regime to ensure the information retrieval ability of the model. Specifically, we set $\gamma = 0.75$, $K = 2048$, $L = 32$, and $B = 4$.

**Relative PE hyperparameters**    For the decay mask, we set $m = -\log(0.8) \approx 0.223$. For RoPE, we set $\theta_i = 10000^{-2(i-1)/d}$, as in Su et al. (2023).

**Compute**    We trained all of our models on a Tesla V100 GPU.

**Training details**    In all experiments, we used the AdamW optimizer (Loshchilov & Hutter, 2019) with a learning rate of $10^{-3}$, a weight decay of $10^{-6}$, a batch size of 128, and trained for $100,000$ iterations.

## K. Additional Experimental Results

### K.1. The effect of depth and relative PEs

Here, we present additional experimental results on the effect of attention depth on positional bias. Since models without residual connections become increasingly difficult to train as depth increases, we limit our experiments to depths of 2, 4, and 6. The results are shown in Figure 4. We observe that deeper attention amplifies the bias toward earlier tokens, regardless of the PE used. Furthermore, both decay mask and RoPE introduce distance-based decay effects that increase focus on recent tokens, but the one induced by decay mask is stronger.

### K.2. The effect of residual connections on position bias

In this section, we present additional experimental results on the effect of residual connections on the positional bias exhibited by the model. The results are shown in Figure 5. As discussed in Section 5.1, we observe that with residual

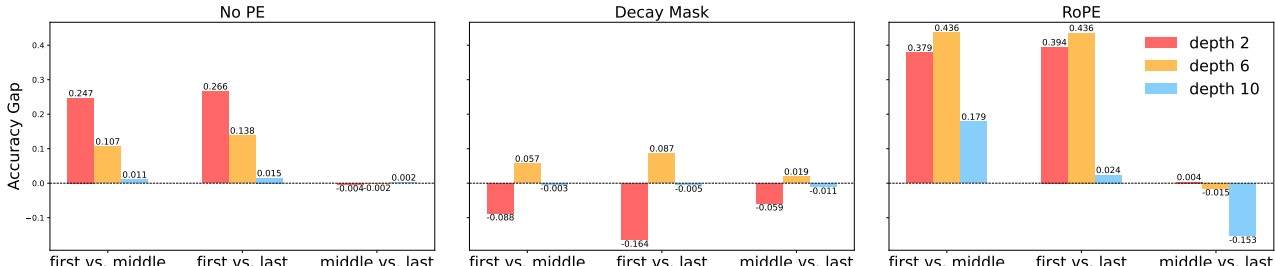

Figure 4: Results with pure attention layers: position bias arising solely from the architectural design of the attention mechanism, with **no positional bias in the training data**.

Figure 5: Results with residual connections: position bias arising solely from the architectural design of the attention mechanism, with **no positional bias in the training data**.

connections, the relationship between model depth and positional bias becomes non-monotonic, and the strength of the bias can be either amplified or reduced depending on the positional encoding used and the depth regime. These findings suggest that residual connections modulate the accumulation of positional effects across layers in a more complex manner than with pure attention alone. It would be interesting to further explore and characterize how residuals influence the emergence and amplification of positional bias.

### K.3. The role of training data on positional bias

In this section, we present additional experimental results building on the experiment described in Section 5.2, but focusing on cases where positional bias in the training sequences is introduced at other positions. Specifically, we consider three types of training sequences where $x_{\text{query}}$ is assigned the class of 1) $x_1$ (the first position), 2) $x_{n/2}$ (the middle position), or 3) $x_n$ (the last position). The corresponding results are shown in Figure 6, Figure 7, and Figure 8, respectively.

Observe that, compared with no mask, the causal mask without PEs indeed introduces a sense of position across all cases. Specifically, it enables the model to learn a positional bias favoring the beginning of the sequence, as earlier tokens tend to receive more attention through the mechanism of iterative attention. In contrast, both sin PE and RoPE allow the model to effectively capture different positional biases regardless of their location in the training sequences.

Interestingly, when comparing this behavior to the case shown in Figure 3, we note that the "lost-in-the-middle" phenomenon only emerges when the training sequences are biased toward both the beginning and the end (the only other exception occurs in the deep-layer regime with RoPE, despite the absence of positional bias in the training data; see Appendix K.2). This suggests that specific types of positional bias in the training data could also play a crucial role in shaping how the model learns to process and prioritize positions within a sequence.

As the structure of positional bias in natural language remains unclear, this observation raises the following question:

*Does positional bias in natural language sequences shape the "lost-in-the-middle" phenomenon in a similar way to what we observe in this simplified case?*

This question connects to broader inquiries about the parallels between artificial and human attention. In neuroscience, the primacy-recency effect highlights that human attention often gravitates toward the beginning and end of sequences (Glanzer & Cunitz, 1966; Li et al., 2024a), a phenomenon that may have influenced the structure of human languages, where critical

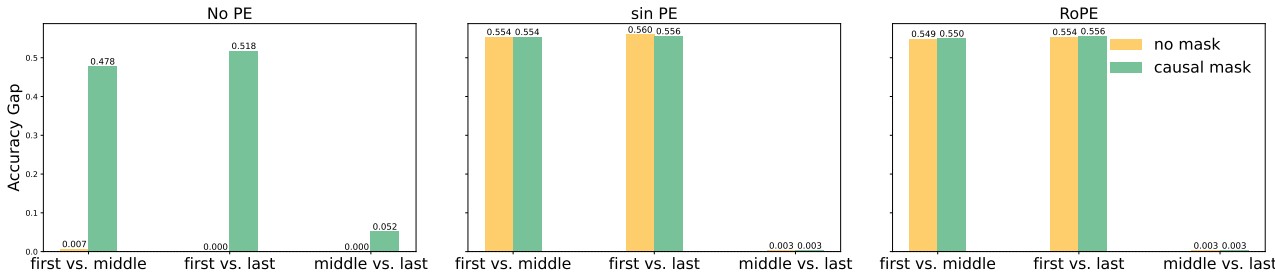

Figure 6: Position bias exhibited by the model when trained on data biased toward the **first** position.

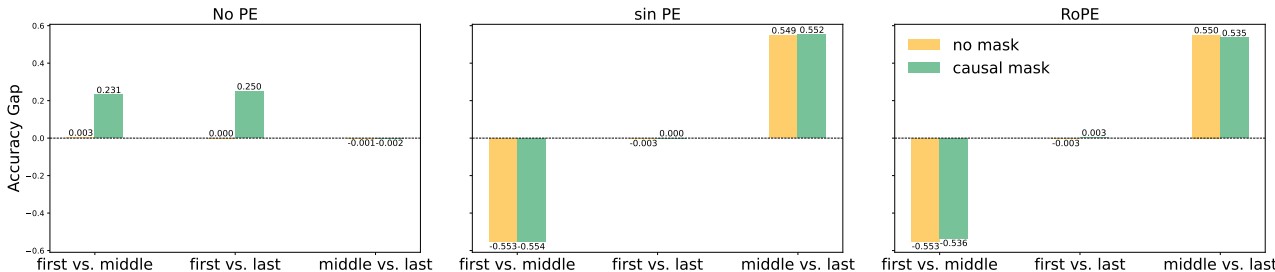

Figure 7: Position bias exhibited by the model when trained on data biased toward the **middle** position.

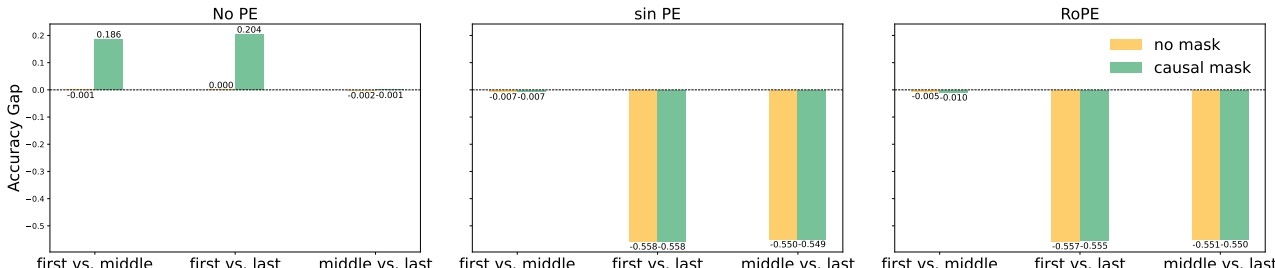

Figure 8: Position bias exhibited by the model when trained on data biased toward the **last** position.

information is frequently placed in these positions (Halliday, 2004). As demonstrated in Section 5.2, when such patterns are present in training data, attention-based architectures seem to develop analogous biases (Hollenstein et al., 2021), aligning with natural language characteristics for improved performance. This raises deeper, perhaps philosophical questions: To what extent are these biases intrinsic to effective sequential processing? How closely should neural networks emulate human cognitive patterns? Investigating these connections can deepen our understanding of both human and artificial intelligence while guiding the design of more effective machine learning models.

### K.4. Attention sinks

Despite our use of a simplified experimental setup in this work, we observe the emergence of key phenomena documented in more complex settings. In addition to the "lost-in-the-middle" phenomenon discussed in Section 5.2 and Appendix K.3, in this section, we report the formation of attention sinks in our setting.

Figure 9 shows an example of the attention maps of a two-layer self-attention networks under the causal mask without PEs, where the sequences used for training and inference are all free of position bias. We observe the similar phenomenon of attention sinks as reported in Xiao et al. (2024).

More quantitatively, following Gu et al. (2025), we calculate their metric for measuring the emergence of attention sinks, over $10,000$ sequences free of position bias. Specifically, denote the adjacency matrix of the mask $\mathcal{G}$ to be $M$. Then the

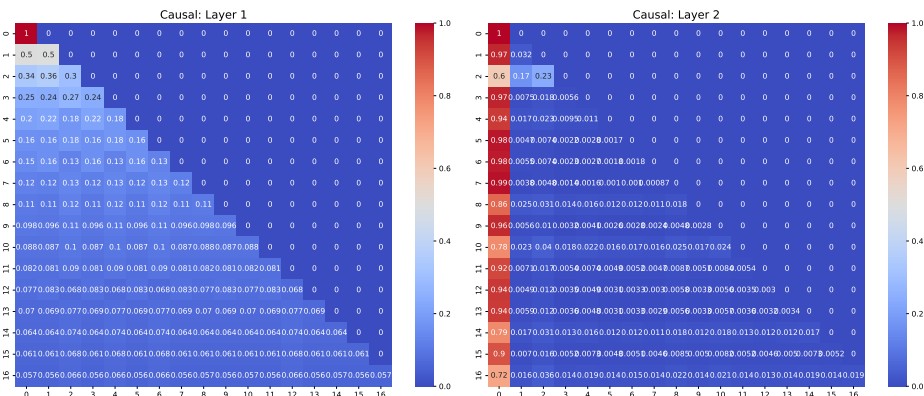

Figure 9: Example of the emergence of attention sinks in our experimental setting. In particular, the sequences used for training and inference are all **free of position bias**.

metric for attention sink at token $j$ is calculated as

$$\text{Attention Sink}_j = \frac{1}{T}\sum_{t=0}^{T-1}\frac{1}{\sum_{i=1}^{N}M_{ij}}\sum_{i=1}^{N}\mathbf{1}\{A_{ij}^{(t)} > \tau\}.$$

The threshold $\tau$ we choose is $0.2$. The results for the causal mask, the sliding-window masks (with $w = 5, 9, 13$), and the prefix masks (with $K = 2, 4, 6$) are shown in Figure 10, Figure 11, and Figure 12, respectively. In particular, we make the following observations:

1. Attention sinks emerge on the absolute first token under the causal mask.

2. Attention sinks tend to emerge on the absolute first token when the window size $w$ is larger, under the sliding-window mask.

3. Attention sinks emerge on the $K$ prefix tokens, not just on the first token alone, under the prefix mask.

All of these phenomena have been observed in real-world LLMs in Gu et al. (2025). This alignment between our controlled setup and real-world observations affirms the validity of our abstraction, indicating that we have captured the key mechanisms underlying position bias while facilitating a systematic analysis.

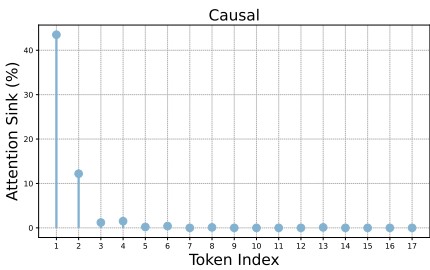

Figure 10: Attention sinks emerge on the first token under the causal attention mask.

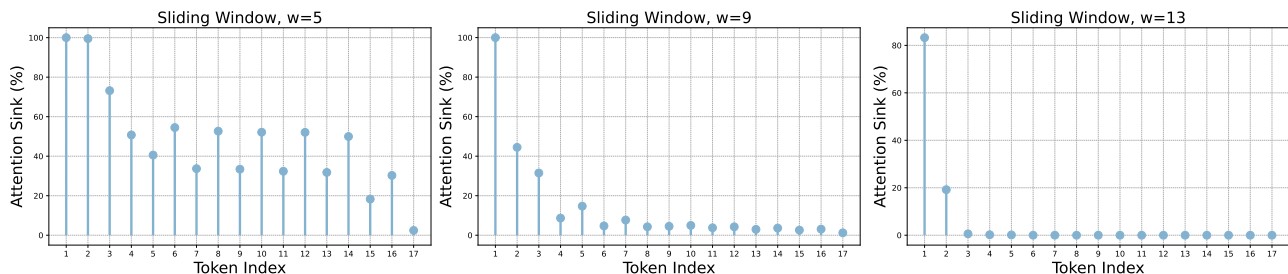

Figure 11: Attention sinks tend to emerge on the absolute first token when the context window size $w$ is larger, under the sliding-window mask.

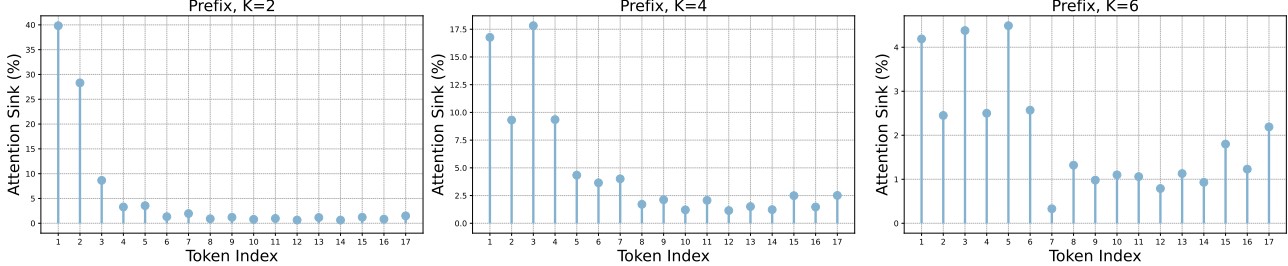

Figure 12: Attention sinks emerge on the $K$ prefix tokens, not just on the first token alone, under the prefix mask.

## L. Additional Results under a Fixed-Vocabulary Setting

In addition to the Gaussian mixture setting discussed in the main text, we also present results for a fixed-vocabulary setting: we define a fixed set of $K$ class vectors $\{v_1, \ldots, v_K\}$, and assign each input $x_i = v_{c(i)}$, where $c(i) \in \{1, \ldots, K\}$ is the class index. Each class vector $v_k$ is then defined by $v_k = (\mu_k + \lambda\omega)/\sqrt{1 + \lambda^2}$, where $\mu_k$ and $\omega$ are $d$-dimensional vectors whose components are sampled $i.i.d.$ from a normal distribution with mean zero and variance $1/d$. Note that the shared component $\eta$ makes $v_k$ not completely orthogonal to each other, resembling token embeddings in practice (Ait-Saada & Nadif, 2023; Cai et al., 2021; Ethayarajh, 2019; Gao et al., 2019; Godey et al., 2024). $\lambda$ is a scalar that controls the strength of anisotropy, where we set $\lambda = 0.75$.

The corresponding results are shown in Figure 13-Figure 24, respectively. Most of the findings presented in the main text continue to hold in this setting. Notably, even with residual connections, increasing depth consistently amplifies positional bias, as shown in Figure 14.

One key difference between the two settings is that, compared to the orthogonal embeddings used in the Gaussian mixture setting of Reddy (2024), the anisotropic embeddings in our setup appear to make the task significantly more challenging.

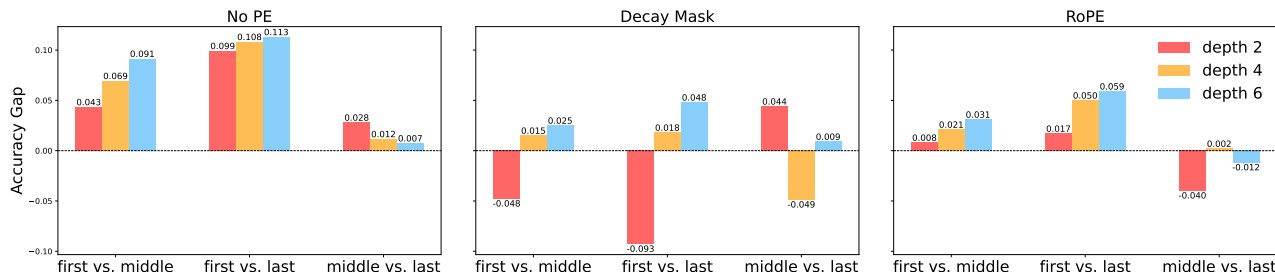

Figure 13: Results with pure attention layers under the fixed-vocabulary setting: position bias arising solely from the architectural design of the attention mechanism, with **no positional bias in the training data**.

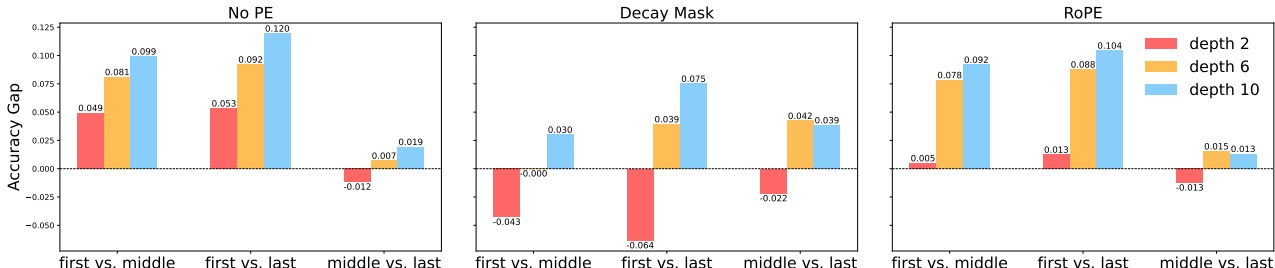

Figure 14: Results with residual connections under the fixed-vocabulary setting: position bias arising solely from the architectural design of the attention mechanism, with **no positional bias in the training data**.

We note that in preliminary experiments under the fixed-vocabulary setting with $\lambda = 0$, the results closely resemble those from the Gaussian mixture setup, suggesting that embedding geometry plays a critical role in task difficulty. It is possible that residual connections behave differently when the task becomes more difficult. Future work could further investigate how the geometry of the input embeddings influences both task difficulty and the emergence of positional bias.

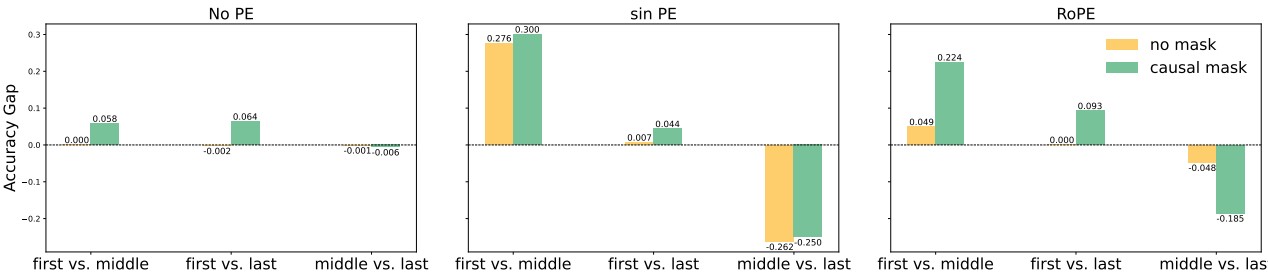

Figure 15: Position bias exhibited by the model when trained on data biased toward the **first and last** positions.

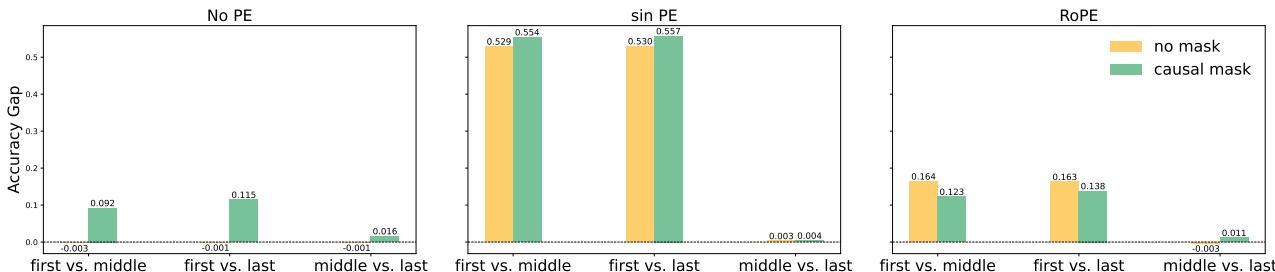

Figure 16: Position bias exhibited by the model when trained on data biased toward the **first** position.

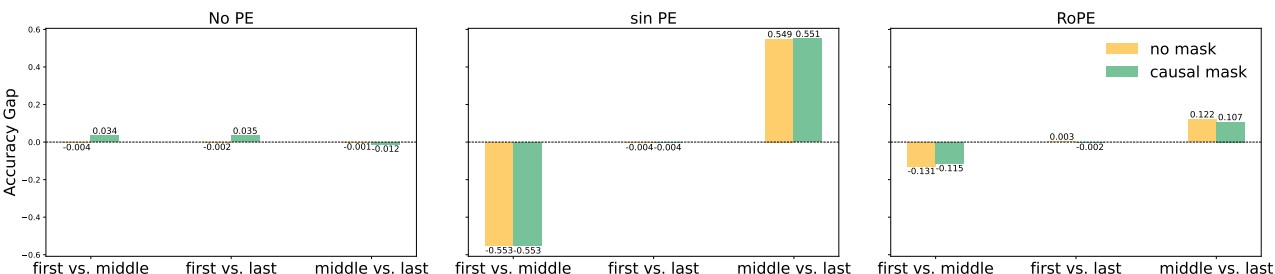

Figure 17: Position bias exhibited by the model when trained on data biased toward the **middle** position.

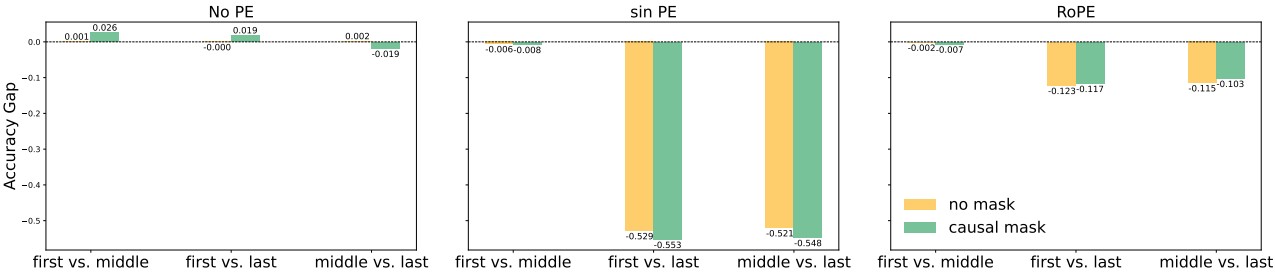

Figure 18: Position bias exhibited by the model when trained on data biased toward the **last** position.

Figure 19: Results under fixed-vocabulary setting

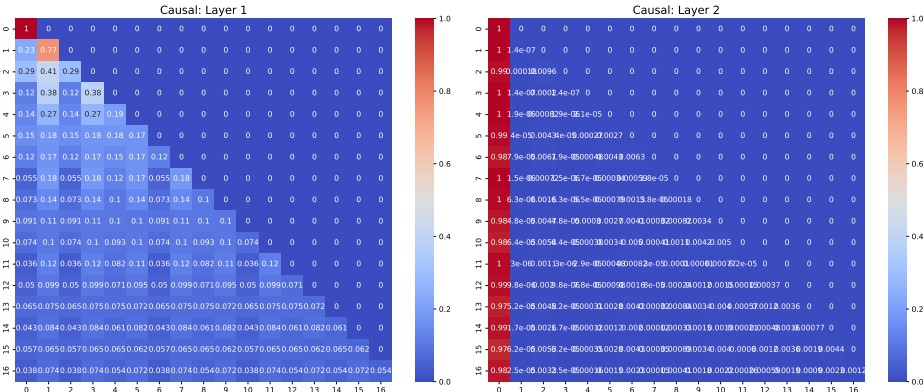

Figure 20: Fixed-vocabulary setting: example of the emergence of attention sinks in our experimental setting. In particular, the sequences used for training and inference are all **free of position bias**.

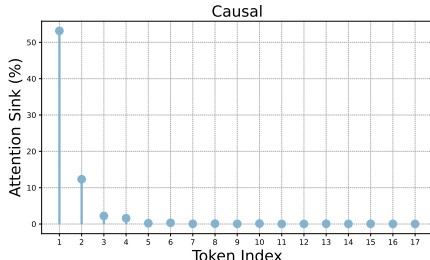

Figure 21: Attention sinks emerge on the first token under the causal attention mask.

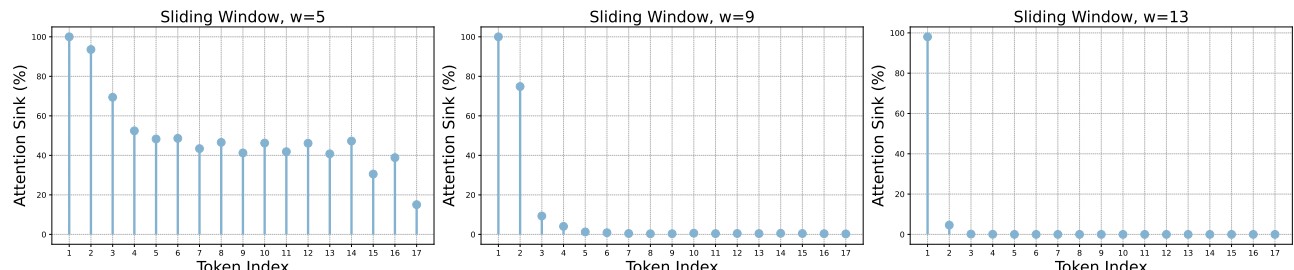

Figure 22: Attention sinks tend to emerge on the absolute first token when the context window size $w$ is larger, under the sliding-window mask.

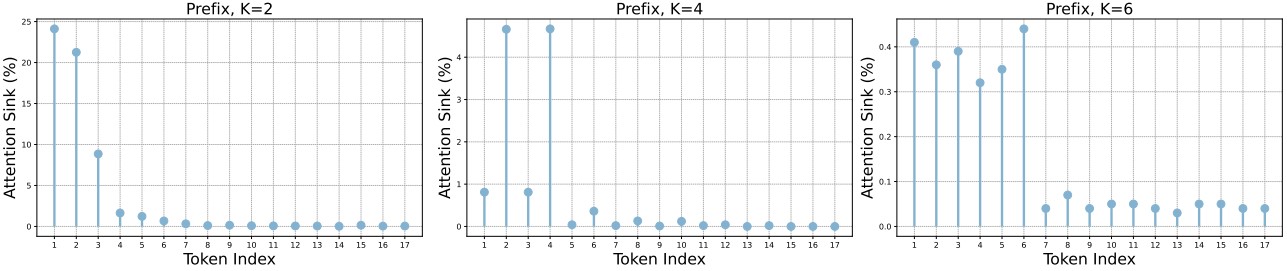

Figure 23: Attention sinks emerge on the $K$ prefix tokens, not just on the first token alone, under the prefix mask.

Figure 24: Attention sinks under the fixed-vocabulary setting.

