# OpenReview forum: "On the Emergence of Position Bias in Transformers"
_ICML.cc/2025/Conference — ICML 2025 poster_

### Official Review · Reviewer_qhRD · 2025-03-09

**Overall Recommendation:** 5

**Summary:**

This paper studies the "position bias" in transformers, that is, the bias of the model to focus on certain regions of the input. The authors investigate how causal mask and positional encoding impact this position bias. To that end, they leverage a graph-theoretic formalization of the attention module to study the position bias. In particular, the authors unify under their study this bias and its empirical observations shown in prior works. The authors show that causal masking biases attention towards earlier positions and conduct this analysis for several types of masking used in practice. The authors also study how positional encoding and masking interact and provide insights into how to design them to trade-off between local and global context. Finally, the authors validate their findings experimentally in a controlled setting.

**Claims And Evidence:**

The theoretical claims are well supported by clear and detailed proofs, and the authors also provide experimental validation of their theory in a controlled setting.

**Essential References Not Discussed:**

To the best of my knowledge, there are no essential references not discussed in the current submission.

**Experimental Designs Or Analyses:**

The authors provided the code to reproduce their results. A quick inspection of the code seems to indicate that the experiments were well-conducted, although I did not re-run the experiments. The experimental design is sound, and results are analyzed, although the experiments are only done in a controlled, hence not very realistic, setting.

**Methods And Evaluation Criteria:**

The authors provide theoretical results to better understand position bias in attention blocks. They conduct experiments to validate their findings. I believe the method and evaluation criteria make sense for the problem at hand.

**Other Comments Or Suggestions:**

None

**Other Strengths And Weaknesses:**

**Strenghts**

- The paper is very well written with detailed prior work, clear notations and technical background
- I find the graph formalization very elegant and innovative.
- The theory is sound, with theorems contextualized and their implications explained
- The contributions summarized in Table 1 are impressive, especially given the many connections to empirical findings in prior works

**Weaknesses**

I list below what I think are weaknesses, but I would be happy to be corrected if I misunderstood some important aspects of the authors' contributions.
- The current analysis focuses on self-attention-only networks, while transformers include feed-forward layers. Although tokens are only interconnected in the attention blocks, the influence of feed-forward layers is not negligible, and I wonder how it would impact the current analysis. Could the authors elaborate on this point?

- In the experiments, the network and data considered are simple, which, I believe, is due to the need to be in a controlled setting to investigate properly the position bias. Could the authors elaborate on larger-scale experiments that could validate their theory or that can benefit from their findings (I do not ask the authors to conduct them, but rather to add a discussion for future work in more practical settings)? For instance, this could be in a limitation or discussion section at the end of the paper.

- Given that the graph-theoretic framework of attention was introduced in [1], I think that the mention of "novel graph-theoretic framework" in the abstract is not adapted. I acknowledge that this does not reduce the novelty of the contribution since it is the first time it is used to study the position bias, but I would appreciate if the author could elaborate on this if I misunderstood something or remove this term in the abstract if the framework is indeed not new.

Overall, I find the paper interesting and the analysis well conducted. Although the model is simplified and the experiments are limited to controlled settings, I believe this is valuable work to better understand position bias in transformers. This is the reason for my current score and I would appreciate if authors could clarify the points mentioned above.

**Update after rebuttal**: I increased my score from 4 to 5.

**Questions For Authors:**

1) Could the authors elaborate on how the analysis conducted in the current submission could be extended or influenced by considering feed-forward layers? In particular, does the position bias only depend on the attention layers since this is the only block where tokens are interconnected, or can the MLPs also influence it, maybe in some indirect manner?

2) In the paragraph after Theorem 1, the authors discuss the role of softmax. If I understood correctly, softmax cannot lead to disconnected graph components since it is not sparse and cannot have exactly zero entries. The authors mention that, empirically, replacing softmax with ReLU can mitigate the emergence of attention sinks. What do the authors think of other families of "sparse" softmax, like sparsemax [1] or fenchel-young derived softmax [2]. From Theorem 1, would it make sense to study them through the lens of avoiding attention sinks?

3) Could the author add a discussion/limitation section regarding the scope of their study (controlled setting, simplified transformer) and some room for future work in more practical settings?

*References*

[1] Martins et al. From Softmax to Sparsemax: A Sparse Model of Attention and Multi-Label Classification, ICML 2016

[2] Blondel et al. Learning with Fenchel-Young Losses, JMLR 2020

**Relation To Broader Scientific Literature:**

I find that related work and prior works are well introduced and compared. Although the graph-formalization was introduced in a prior work [1], the submission's contributions seem to be novel regarding the understanding of position bias in attention layers by leveraging the graph formalization of attention layers.

*References*

[1] Wu et al. On the Role of Attention Masks and LayerNorm in Transformers, NeurIPS 2024

**Theoretical Claims:**

The proofs are well-detailed and clear.

---

> ### Author Rebuttal · Authors · 2025-04-01
>
> Thank you for your thoughtful comments and positive assessment of our work. Below, we provide individual responses to the comments and questions you raised.
>
> > I wonder how MLPs would impact the current analysis.
>
> Thank you for the question. Our analysis focuses on the self-attention mechanism, as it is the primary component responsible for information exchange across tokens. In contrast, MLP layers are typically applied independently to each token and do not directly contribute to inter-token communication. As such, they are unlikely to affect position bias in the same way that attention mechanisms do.
>
> That said, we acknowledge that MLPs, being universal function approximators, can theoretically induce position-dependent effects under specific conditions. For instance, one could construct a scenario where an MLP maps $x_i$ to $x_{n-i+1}$ for a sequence  $x_1, x_2, x_3, x_4, …, x_{n-1}, x_n$​, thereby indirectly altering token interactions. However, such behaviors would require highly specific weight configurations that are unlikely to arise under standard training, and they would generally lack robustness to input variation.
>
> Our goal in this work is to isolate and understand the role of attention masks and positional encodings in shaping position bias—factors that more directly govern how positional information is integrated across tokens. Nonetheless, extending our framework to investigate whether and how MLP layers might interact with or amplify these biases is a valuable direction for future work, and we will add a discussion of this point in the revised manuscript.
>
> > What do the authors think of other families of "sparse" softmax, like sparsemax or fenchel-young derived softmax. Would it make sense to study them through the lens of avoiding attention sinks?
>
> Thank you for the question. Our current findings suggest that sparser attention mechanisms may help mitigate attention sinks by slowing the convergence of attention flow. In our framework, this is captured by Theorem 4.2, which shows that increased sparsity (e.g., shorter sliding windows) causes center nodes to accumulate influence more gradually, making them less likely to dominate early on. Consistently, we observe empirically that attention sinks are less likely to emerge under sparser masking conditions (Figure 10).
>
> This connection suggests that sparse alternatives to softmax could act as implicit regularizers, dampening the formation of attention sinks and promoting more balanced information flow across tokens. Extending our analysis to formally validate and characterize these effects would be a valuable direction for future work, and we will note this in the revised manuscript.
>
> > Could the authors elaborate on larger-scale experiments that could validate their theory or that can benefit from their findings?
>
> Thank you for the question. A key contribution of our theoretical framework is that it helps identify and quantify the sources of attention bias introduced by masking and positional encodings. These insights can inform the design of alternative transformer architectures, by motivating positional encoding schemes or masking strategies that encourage more uniform or task-aligned attention distributions.
>
> Larger-scale experiments could build on our findings to evaluate how different architectural choices impact position bias and downstream performance on different tasks. Conversely, one could also explore whether alignment between a model’s learned position bias and natural language structure (e.g., recency effects in Futrell et al.) correlates with improved language modeling performance.
>
> > Given that the graph-theoretic framework of attention was introduced in Wu et al. (2024), I think that the mention of "novel graph-theoretic framework" in the abstract is not adapted.
>
> Thank you for the thoughtful comment. We agree that the use of graph-theoretic tools to analyze attention mechanisms has been explored in prior work, notably by Abnar et al. and Wu et al. We will revise the abstract to more accurately reflect our contribution. We will revise the wording in the abstract to more precisely reflect our contribution.
>
> The novelty of our work lies not in the use of graphs itself, but in how we apply this perspective to analyze the flow of information and the emergence of positional bias across multiple layers of attention. Our framework leverages the structure induced by causal and other attention masks to provide theoretical insight into how positional preferences arise independent of semantic content. We will clarify this distinction in the revised abstract to better situate our contributions.
>
> We sincerely appreciate your feedback and welcome any further suggestions.
>
> ---
> References
>
> Abnar et al. (2020) Quantifying Attention Flow in Transformers.
>
> Futrell et al. (2015) Large-scale evidence of dependency length minimization in 37 languages.

---

> > ### Comment · Reviewer_qhRD · 2025-04-03
> >
> > I thank the author for the rebuttal that addresses my concerns. I maintain my evaluation: this is a clear and very well-presented work with valuable contributions to better understanding attention-based models. As discussed in the rebuttal above, it also opens interesting questions for analysis on larger models (theoretically or experimentally). Although there are areas of improvement, in my humble opinion, such papers are valuable to the community and should be published. To highlight that, I increased my score to 5.

---

### Official Review · Reviewer_hruX · 2025-03-11

**Overall Recommendation:** 4

**Summary:**

This paper presents a graph-theoretic framework to analyze how position bias emerges in transformer architectures. The authors mathematically model attention masks as directed graphs to understand how tokens interact based on their sequential positions across multiple layers of attention.
The authors support their findings with experiments that reproduce position biases observed in real-world LLMs, including phenomena like "lost-in-the-middle" and attention sinks. Their framework helps explain why increasing model depth amplifies positional biases, why some positions gain disproportionate attention, and how different masking approaches affect information flow.

The main claims of the paper are:
- Causal masking inherently biases attention toward earlier positions in deep networks. This happens because tokens in deeper layers attend to increasingly contextualized representations of earlier tokens, amplifying their influence.
- There's a nuanced interaction between causal masking and relative positional encodings (like decay masks and RoPE). While these encodings introduce distance-based decay within individual attention maps, their aggregate effect across multiple layers creates a trade-off between local decay effects and the cumulative importance of early sequence positions.

**Claims And Evidence:**

The connection between center nodes in their graph-theoretic framework and attention sinks in real models is theoretically elegant but would benefit from more direct empirical validation with actual LLM attention patterns rather than just the controlled experiments.

**Essential References Not Discussed:**

I'd suggest the authors to check these papers:

- Stolfo, Alessandro, et al. "Confidence regulation neurons in language models." Advances in Neural Information Processing Systems 37 (2024): 125019-125049.

- Cancedda, Nicola. "Spectral filters, dark signals, and attention sinks." arXiv preprint arXiv:2402.09221 (2024).

**Experimental Designs Or Analyses:**

The experimental design is generally sound but has some minor limitations:

- The experiment uses very small model sizes (n=8, d=64) compared to real LLMs, raising questions about whether the observed effects scale appropriately to larger models.
- While the authors test three different position encoding schemes, they use only a single implementation of each. Testing variants (like, different decay rates for decay masks or base frequencies for RoPE) would strengthen the generalizability of the findings.
- The statistical analysis is minimal, with results presented as averages over five runs but without confidence intervals or significance testing, making it difficult to assess the reliability of the observed effects.

**Methods And Evaluation Criteria:**

The authors' methodological choices seem good:

- Using a graph-theoretic framework is appropriate for analyzing information flow in attention mechanisms.
- The controlled experimental setting from Reddy (2024) allows for isolating the effects of different architectural components on position bias.
- The evaluation metric (accuracy gap between different positions) directly measures position-dependent performance differences.

Even though, focusing only on syntetic data leaves the question on whether the same dynamics would emerge in natural language data with more complex semantic structures.

**Other Comments Or Suggestions:**

N/A

**Other Strengths And Weaknesses:**

other weaknesses:
- The paper primarily analyzes the attention mechanism without deeply exploring how feed-forward networks, residual connections, and other transformer components might interact with position bias.

other streanghts:
- The work successfully connects multiple empirical observations about position bias (attention sinks, lost-in-the-middle effect) within a coherent theoretical framework, offering deeper understanding of these phenomena.
- The authors provide formal proofs for their theorems, making their claims verifiable and establishing a solid foundation for future theoretical work in this area.
- The controlled experiments are well-designed to isolate the effects of different components (mask type, positional encoding, model depth) on position bias, validating the theoretical findings.

**Questions For Authors:**

N/A

**Relation To Broader Scientific Literature:**

I think this paper is valuable: the paper builds a bridge between empirical observations of positional phenomena in transformers and theoretical understanding of their architectural causes, while in the literature these phenomena were observed but not fully explained.

- The paper connects to the literature on the "lost-in-the-middle" phenomenon described by Liu et al. (2024), Zhang et al. (2024), and Guo & Vosoughi (2024) - the authors provide a theoretical explanation based on the interplay between positional biases from the architecture and biases in the training data, showing how specific patterns of position-dependent performance emerge under different conditions.

- The paper connects to recent work on attention sinks by Gu et al. (2025) and Xiao et al. (2024) - the authors provide a theoretical explanation by showing that attention sinks naturally emerge at center nodes in the directed graph defined by the attention mask. This explains why sinks form at the beginning of sequences under causal masking or at all prefix tokens under prefix masking.

**Theoretical Claims:**

The theoretical claims seem sound.

---

> ### Author Rebuttal · Authors · 2025-04-01
>
> We appreciate your positive assessment and constructive comments, which have helped strengthen our work. Below, we provide responses to your comments.
>
> > The paper primarily analyzes the attention mechanism without deeply exploring how other transformer components might interact with position bias.
>
> Thank you for your thoughtful comment.  We agree that these other transformer components could contribute to position bias in an nontrivial way. We view this as an important direction for future work.
>
> That said, our focus on the attention mechanism in this work is aligned with a common practice in the theoretical analysis of transformers: isolating the attention mechanism to better understand its intrinsic inductive biases. Many prior studies have adopted similar simplifications by omitting or abstracting away certain components in order to enable precise analysis [1-5].
>
> Nevertheless, our analysis can be naturally extended to account for residual connections. Following the renormalization technique in [6], we can redefine the attention matrix at layer t as $A^{(t)}\_{res} = 0.5 A^{(t)} + 0.5I$. This ensures $A^{(t)}_{res}$ remains a valid stochastic matrix and thus retains interpretability.   Under this adjustment, our theoretical results still hold; but the convergence rate would slow down, aligning with findings in [1] that residual connections slow down rank collapse rate under attention.  We will add a remark to the paper detailing how to handle residual connections in our analysis and their effect.
>
> > ...would benefit from more direct empirical validation with actual LLM attention patterns rather than just the controlled experiments.
>
> Thank you for the comment. Natural language lacks ground-truth annotations for position bias, making it difficult to disentangle architectural effects from semantic content. To enable precise control, we adopt the synthetic setup from Reddy, which allows us to study positional bias in isolation. Despite this abstraction, our setup reproduces key behaviors observed in real LLMs, such as the “lost-in-the-middle” effect (Sec. 5.2) and attention sinks (App. K.2). Tab 1 further shows alignment between our results and empirical observations on position bias reported in the literature.
>
> We agree that validating these results directly in real models with natural language data is an important next step. As noted in App K, one direction is to quantify position bias in LLMs and relate it to known linguistic structures [7]. We will highlight these directions in the updated version.
>
>
> > The experimental design has some minor limitations.
>
> Thank you for your comment. The choices n = 8 and d=64 follow from Reddy. In line with your suggestion, we have added more variants for the decay masks and RoPE for the experiment in Sec 5.1 under the same setup:
>
> Decay
> | m | depth | first vs. middle | first vs. last | middle vs. last |
> |---|---|---|---|---|
> |  | 2 | -.025 | -.092 | -.070 |
> | 0.511 | 6 | -.057 | -.059 | -.006 |
> |  | 10 | -.044 | -.049 | -.002 |
> |  | 2 | -.043 | -.064 | -.022 |
> | 0.223 | 6 | .000 | .039 | .042 |
> |  | 10 | .030 | .075 | .039 |
> |  | 2 | .010 | .009 | -.011 |
> | 0.105 | 6 | .079 | .121 | .044 |
> |  | 10 | .110 | .148 | .073 |
>
> RoPE
>
> | $\theta$| depth | first vs. middle | first vs. last | middle vs. last |
> |---|---:|---:|---|---|
> |  | 2 | .005 | .001 | -.002 |
> | 1/100 | 6 | .051 | .070 | .008 |
> |  | 10 | .075 | .088 | .012 |
> |  | 2 | 0.006 | .002 | -0.009 |
> | 1/1000 | 6 | .064 | 0.84 | .015 |
> |  | 10 | .079 | .086 | .018 |
> |  | 2 | .005 | .013 | -0.013 |
> | 1/10000 | 6 | .078 | .088 | .015 |
> |  | 10 | .092 | .104 | .013 |
>
> The results align with our findings in Sec 4, with greater $m$ or $\theta$ inducing greater decay, while deeper attention amplifying the bias toward earlier tokens.
>
> We also present here the standard deviations for the results in Fig 2:
>
> |  |  | first vs. middle | first vs. last | middle vs. last |
> |---|---|---|---|---|
> |   | 2 | 0.016 | 0.014 | 0.009 |
> | no PE | 6 | 0.026 | 0.013 | 0.021 |
> |  | 10 | 0.011 | 0.013 | 0.005 |
> |  | 2 | 0.016 | 0.015 | 0.007 |
> | decay mask | 6 | 0.017 | 0.018 | 0.028 |
> |  | 10 | 0.010 | 0.019 | 0.011 |
> |  | 2 | 0.016 | 0.017 | 0.010 |
> | RoPE | 6 | 0.025 | 0.027 | 0.010 |
> |  | 10 | 0.020 | 0.025 | 0.005 |
>
>
> The standard deviations for Fig 3 show a similar trend that they are small compared to the averages. This supports the robustness of our reported trends.
>
>
> We sincerely appreciate your feedback and welcome any further suggestions.
>
> ----
> References
>
> 1. Attention is not all you need: pure attention loses rank doubly exponentially with depth
>
> 2. Signal propagation in transformers: Theoretical perspectives and the role of rank collapse
>
> 3. A mathematical perspective on transformers
>
> 4. The emergence of clusters in self-attention dynamics
>
> 5. On the Role of Attention Masks and LayerNorm in Transformers
>
> 6. Quantifying Attention Flow in Transformers
>
> 7. Large-scale evidence of dependency length minimization in 37 languages

---

> > ### Comment · Reviewer_hruX · 2025-04-05
> >
> > I thank the authors for providing these clarifications and i'd like to keep my overall score to 4.

---

### Official Review · Reviewer_SCvk · 2025-03-13

**Overall Recommendation:** 4

**Summary:**

This paper analyses position bias in transformers, both theoretically and experimentally.
The paper first proposes to analyse a transformer as a graph, with attention weights representing weighted-edges between two tokens’ representations in adjacent transformer layers; an *attention flow* can then be computed between tokens $t$ and $t’$ at layers $\ell$ and $\ell’$, respectively, by summing the attention over all possible paths between them.
The paper then uses this graph-based framework to show how the attention flow changes as the depth of a transformers grows to infinity.
They consider models with autoregressive, sliding, and prefix attention masks.
Importantly, they show that for these three models all attention flow converges to the first token position as the depth of a model grows.
They also consider models with no position embeddings, with ALiBi, and with ROPe.
They then experiment on a synthetic task, with results supporting their theoretical analysis.

# Strengths

This paper studies an interesting phenomenon—position bias in autoregressive language models—providing new theoretical results justifying their existence.

This paper then supports this theoretical analysis with well designed experiments.

# Weaknesses

The paper misses some critical literature in interpretability and analysis of language models, which could make analyses stronger if considered. In particular, the paper fails to cite Abnar et al. (2020), who first proposed the graph-theoretical analysis of transformers used in this paper (termed attention flow there). This framework has later been criticised/expanded by, e.g., Kobayashi et al. (2022), who point out that considering value-vector norms is important for more meaningful analyses involving attention flow. Further, discussing papers such as Jain et al. (2019) and Wiegreffe et al. (2019), and their relation to the analyses proposed here could significantly strengthen the paper.

The theoretical analysis ignores residual connections, which are integral to modern transformers. Residual connections can be interpreted as increasing the effect attention by one on diagonal entries, i.e.,: $A = \widehat{A} + I$. Given this extra term, the condition necessary to prove Theorem 4.1 (i.e., $P_{ij} < 1- \epsilon$) does not hold. So, do the theoretical analyses here only hold for transformers without residual connections? Furthermore, were residual connections used in experiments?

* Jain et al. (2019). Attention is not Explanation. https://aclanthology.org/N19-1357.pdf
* Wiegreffe et al. (2019). Attention is not not Explanation. https://aclanthology.org/D19-1002.pdf
* Abnar et al. (2020). Quantifying Attention Flow in Transformers. https://aclanthology.org/2020.acl-main.385.pdf
* Kobayashi et al. (2022). Attention is Not Only a Weight: Analyzing Transformers with Vector Norms. https://aclanthology.org/2020.emnlp-main.574.pdf

**Claims And Evidence:**

Yes. The paper supports its claims with both theoretical analyses and practical experiments.

**Essential References Not Discussed:**

Yes. The paper claims one of its contributions is proposing a new graph-based framework for analysing transformers, but this framework was already proposed by Abnar et al. (2020). In particular, the main value analysed in this paper $P_{ij}^{(t)}$ is what Abnar et al. (2020) term attention flow.

Abnar et al. (2020). Quantifying Attention Flow in Transformers. https://aclanthology.org/2020.acl-main.385.pdf

**Experimental Designs Or Analyses:**

All experiments seemed sound and valid to support the paper's claims.

**Methods And Evaluation Criteria:**

Yes. The only issue in this regard is the lack of residual connections in the theoretical analyses.

**Other Comments Or Suggestions:**

* Line 125: Should matrices W_q and W_k have size d_{QK} instead of d’? The same question for W_v in line 146.
* Attention sink and center node: As I understand it, attention sinks are not about the limiting behavior of attention flow as $t \to \infty$. Attention sinks are typically defined as a specific (single) attention head which puts all attention mass on a single previous position. If that’s the case, then your analysis does not actually justify the existence of attention sinks, right? Or does it?

**Other Strengths And Weaknesses:**

N/A

**Questions For Authors:**

N/A

**Relation To Broader Scientific Literature:**

I believe the lack of connection with prior work is the largest issue in this submission. The paper misses some critical literature. In particular, the paper fails to cite Abnar et al. (2020), who first proposed the graph-theoretical analysis of transformers used in this paper (termed attention flow there). This framework has later been criticised/expanded by, e.g., Kobayashi et al. (2022), who point out that considering value-vector norms is important for more meaningful analyses involving attention flow. Further, discussing papers such as Jain et al. (2019) and Wiegreffe et al. (2019), and their relation to the analyses proposed here could significantly strengthen the paper.

* Jain et al. (2019). Attention is not Explanation. https://aclanthology.org/N19-1357.pdf
* Wiegreffe et al. (2019). Attention is not not Explanation. https://aclanthology.org/D19-1002.pdf
* Abnar et al. (2020). Quantifying Attention Flow in Transformers. https://aclanthology.org/2020.acl-main.385.pdf
* Kobayashi et al. (2022). Attention is Not Only a Weight: Analyzing Transformers with Vector Norms. https://aclanthology.org/2020.emnlp-main.574.pdf

**Theoretical Claims:**

I checked the two first proofs and did not immediately find any issues.

---

> ### Author Rebuttal · Authors · 2025-04-01
>
> We appreciate your positive assessment and constructive feedback, which have helped strengthen our work. Below, we provide individual responses to the comments you raised.
>
> > The paper misses some critical literature in interpretability and analysis of language models.
>
> Thank you for the pointers to these important references. In particular, we will clarify how our approach aligns with, yet differs from, the “attention rollout” and “attention flow” methods proposed by Abnar et al. (2020). While those methods conduct post-hoc graph-based analyses on specific inputs to understand information flow, our work takes a more general and theoretical perspective: we formalize attention masks themselves as directed graphs (Sec. 3) and analyze how their structure shapes the flow of information across layers—independent of semantic content.
>
> This allows us to rigorously trace how positional information is integrated across general sequences, highlighting the inductive bias introduced by masking alone. We hope this theoretical framework—which connects architectural design to emergent behavior—offers a useful new tool for understanding and probing attention mechanisms in future research.
>
> We appreciate your feedback and will work diligently to ensure that our final version properly suits our contribution within the broader literature on understanding attention.
>
> > The theoretical analysis ignores residual connections, which are integral to modern transformers.
>
> Thank you for your comment. One way to incorporate residual connections into our analysis is to follow the renormalization approach used in Abnar et al. (2020), where each attention matrix $A^{(t)}$ is adjusted to $A^{(t)}\_{res} = 0.5 A^{(t)} + 0.5I$. This ensures $A^{(t)}_{res}$ remains a valid stochastic matrix and thus retains interpretability.   Under this adjustment, our theoretical results still hold; but the convergence rate would slow down, aligning with results in Dong et al. (2021) showing that residual connections slow down the rank collapse rate in token representations.  We will add a remark to the final manuscript detailing how to handle residual connections in our analysis and their effect.
>
> > Should matrices $W_q$ and $W_k$ have size $d_{QK}$ instead of $d'$?
>
> Good catch! Indeed, in many practical implementations, $d'$ matches $d_{QK}$​, However, we keep them distinct in our formulation to accommodate more general scenarios where these dimensions may differ (e.g.  Wang et al. (2024) proposes adjusting $d_{QK}$​​ at inference time to handle longer contexts more effectively).
>
> > Attention sink and center node: do your analysis does not actually justify the existence of attention sinks, right? Or does it?
>
> Thank you for the thoughtful question. Our analysis does not directly justify the existence of attention sinks—defined as single-layer heads that concentrate all attention on a single token. Rather, our goal is to provide structural insight into where attention sinks tend to emerge and why they become more prominent in deeper layers.
>
> - In particular, we observe that the positions where attention sinks are most likely to appear (e.g., the first token under causal and sliding window masks, or the first K tokens under prefix masks; see Xiao et al., (2024); Gu et al., (2025)) coincide with the center nodes in the attention-mask graph defined by our framework. These nodes act as structural attractors in the multi-layer attention dynamics, helping to explain why they accumulate disproportionate influence over time.
>
> - Thus, while attention sinks remain a per-layer phenomenon, our graph-theoretic perspective offers a complementary explanation for their emergence in specific positions and their amplification in deeper layers.
>
> We appreciate your questions and comments very much. Please let us know if you have any further questions.
>
>
> ----
> References
>
> Wang et al. (2024)  Length Generalization of Causal Transformers without Position Encoding.
>
> Xiao et al. (2024) Efficient Streaming Language Models with Attention Sinks.
>
> Gu et al. (2025) When Attention Sink Emerges in Language Models: An Empirical View.
>
> Dong et al. (2021) Attention is Not All You Need: Pure Attention Loses Rank Doubly Exponentially with Depth.
>
> Abnar et al. (2020). Quantifying Attention Flow in Transformers.

---

> > ### Comment · Reviewer_SCvk · 2025-04-03
> >
> > I thank the authors for their response. However, their response regarding attention rollouts of Abnar et al. (2020) does not clarify any difference between both frameworks to me. In particular, you claim:
> >
> > > While those methods [Abnar et al.'s] conduct post-hoc graph-based analyses on specific inputs to understand information flow, our work takes a more general and theoretical perspective: we formalize attention masks themselves as directed graphs (Sec. 3) and analyze how their structure shapes the flow of information across layers—independent of semantic content.
> >
> > I still do not see any differences between your graph-formalism, and the one from Abnar et al. (2020). In fact, I am convinced that the values $\mathbb{P}^{(t)}(z_i = j \mid X^{(0)})$ you analyse are mathematically equivalent to attention rollouts. (In my review above I said attention flow, but it's actually the rollouts which are equivalent. I apologise for the confusion between the two names.) If they are not, could you please show mathematically how they differ?
> >
> > In regards to "formalising attention masks as directed graphs and normalised attention logits as edge weights", vs. "directly formalising attention weights (computed using the said mask) as edge weights" (is this the point you're making in how frameworks differ?): these two things are, in my opinion, equivalent. As the attention mask forces some attention weights to 0, those edges will not exist in the formalism of Abnar et al., and the analysed graphs will be identical.
> >
> > > This allows us to rigorously trace how positional information is integrated across general sequences, highlighting the inductive bias introduced by masking alone. We hope this theoretical framework—which connects architectural design to emergent behavior—offers a useful new tool for understanding and probing attention mechanisms in future research.
> >
> > Given the two points above, I still think your graph-based framework is identical to Abanar et al.'s and that this "rigorous tracing" could be derived from either. As highlighted in my review above, I don't think this takes away from your paper, it still has plenty of value to warrant a "4" even without the graph-based framework being one of its contributions -- however, I believe it's important to clearly acknowledge the source of this framework as Abnar et al.'s. As such, I am lowering my score. I will raise it again if the authors either: make a more convincing argument for why their framework differs from Abnar et al.'s, or acknowledge that this contribution comes from prior work.
> >
> > ## Regarding the Other Related Work
> >
> > I would appreciate it if the authors clearly discussed the other related work I mentioned in my review in their paper. Either highlighting the limitations of this paper in not adopting, e.g., the advances to the framework presented by Kobayashi et al. (2022), or discussing why this is not an issue with its analysis.
> >
> > ## Regarding Residual Connections
> >
> > Thanks for the response. I agree that adding that to the paper would make it stronger. A short discussion on why you formalise residuals as `.5A + .5 I` instead of `A + I` would also be pertinent in my opinion. I believe the latter would be more compatible with how residuals are actually implemented in transformers. Furthermore, formalising it as `A + I` would invalidate your theorems, right? Or do results hold then? I think this should be highlighted as a limitation in the manuscript.
> >
> > ## Summary
> >
> > Again, I liked this paper and would like to see it accepted (that's why I originally gave it a 4). On the other hand, I do think it is important to clearly acknowledge prior work's contributions (and thus I am lowering my score). I will increase my score again if either I'm convinced of the difference between this framework and Abnar et al.'s, or if the authors acknowledge they are equivalent. If the AC disagrees the frameworks are equivalent, they should disregard this new lower score and use my old score instead (i.e., a score of 4).

---

> > > ### Author Response · Authors · 2025-04-05
> > >
> > > We appreciate your thoughtful evaluation and would like to clarify several points from our previous rebuttal, as well as the novelty and scope of our contributions.
> > >
> > > > I still do not see any differences between your graph-formalism, and the one from Abnar et al. (2020). In fact, I am convinced that the values you analyse are mathematically equivalent to attention rollouts.
> > >
> > > Thank you for pointing this out. We agree that $P(z_i = j | X^{(0)})$ in our work is mathematically equivalent to the attention rollout values $\tilde{A}$ from Abnar et al. (Eq. 1). We were not aware of this prior work when writing the initial draft and appreciate you bringing it to our attention. As described in Sec 3, we derived this quantity independently from the probabilistic interpretation of attention in Kim et al. (2017). We will acknowledge this equivalence and properly attribute Abnar et al. and other works in the revised manuscript.
> > >
> > > > In regards ... these two things are, in my opinion, equivalent. Given the two points above, I still think your graph-based framework is identical to Abnar et al.'s and that this "rigorous tracing" could be derived from either.
> > >
> > > Thank you for the comment. As discussed above, we agree that the two formulations are equivalent and we will acknowledge that in our revision. That said, we want to clarify and highlight two main distinctions between ours and Abnar et al.’s work:
> > >
> > > 1. Goal: Rather than claiming novelty in the graph abstraction itself (we apologize for any confusion caused by our earlier wording, which we will change), we leverage this powerful graph view, pioneered by Abnar et al., to study how architectural choice (masks, PEs, model depth) shapes the way how attention integrates positional information independent of specific input semantics.
> > > - This is an important distinction with Abnar et al.. A central challenge in analyzing attention mechanisms lies in **disentangling the effects of semantics and position**, as attention outputs are influenced by both. Empirical approaches like attention rollout (as used in Abnar et al.) are highly valuable for tracing information flow on specific inputs. However, they do not directly yield general analytical insights that hold across all inputs.
> > >
> > > 2. Methodology and theoretical significance: when we refer to our "graph-theoretic framework," we mean the suite of **graph theory based proof techniques** we develop (such as walk counting, dynamic programming, and graph compression) which enable exact enumeration and analysis of attention paths for general inputs. These techniques enable us to prove **non-asymptotic, input-agnostic theoretical results with explicit convergence rates** that characterize how attention mechanisms propagate information over **arbitrary** sequences.
> > >
> > > - While the graphs themselves are structurally the same to those in Abnar et al., our analysis focuses on deriving **quantitative and provable statements about model inductive bias**—insights that empirical attention rollout computation alone cannot provide.
> > >
> > > We will clarify these distinctions and explicitly acknowledge the contributions of Abnar et al. and other related works, including the attention rollout formulation and the use of graphs for visualizing information flow. We hope this helps clarify how our contributions differ in motivation, methodology, and theoretical scope, and how our framework builds on and complements existing literature.
> > >
> > > ## Other related work
> > >
> > > Thank you for pointing us to these important references. We will add a dedicated section in the related work to discuss them.
> > >
> > > - Our analysis focuses on attention weights, as they are the primary component for inter-token communication. In contrast, value projections and MLPs are typically applied independently to each token and are unlikely to affect position bias in the same way that attention weights do.
> > >
> > > - That said, we agree that under specific conditions, value projections or MLPs can induce position-dependent effects. For example, if a value matrix $V$ maps $x_i$ to $x_{n-i+1}$. However, such behaviors would require highly specific weight configurations that are unlikely to arise under standard training, and they would generally lack robustness to input variation.
> > >
> > > - Our goal is to isolate the effects of attention masks and positional encodings, which more directly govern how positional information is integrated. Nevertheless, extending our framework to include potential interactions with value projections and MLPs, as explored in Kobayashi et al., is a promising direction for future work, and we will add this discussion in the revision.
> > >
> > > ## Residual connections
> > >
> > > Formalizing residual connections as $A+I$ would cause $P(z_i = j | X^{(0)})$ to lose its probabilistic interpretation and diverge as $t\to \infty$. Using $0.5A + 0.5I$ ensures the quantity to retain a probabilistic interpretation and well-behaved limiting behavior. We will note this modeling choice and its motivation in the revised manuscript.

---

### Official Review · Reviewer_dVk1 · 2025-03-13

**Overall Recommendation:** 1

**Summary:**

This paper aims to analyze the effect of attention masks, such as the causal mask, of the observed attention patterns. In particular, the authors suggest modeling the possible paths that the information can flow using attention as edges of a graph. By looking at this graph, they obtain certain bounds which they map to practical observations such as bias towards early tokens in the sequence. Furthermore, the paper looks into the effect of relative and rotary positional encodings on these bounds and show how these encodings can lead to different biases.

**Claims And Evidence:**

In addition to the major issue I mention in the next section, some of the claims are not fully justified. For example, theorem 4.1 only yields an upper bound on the weight given to a token. While the upper bound is smaller for later tokens, this doesn't mean that there is necessarily a bias towards earlier tokens as the upper bound can be arbitrarily large and not be strict. Therefore, I find these claims to be misleading.

**Essential References Not Discussed:**

No comments.

**Experimental Designs Or Analyses:**

In Kazemnejad et. al. the authors explicitly show a set of weights for a transformer model that lead to it learning positional indices whereas the paper is claiming this can not be achieved based on experiments. Since the given weights in Kazemnejad et. al. actually allow learning positional indices, I believe these results require further investigation to understand what is different.

**Methods And Evaluation Criteria:**

Based on my understanding, the graph model that is being used, looks at the weight given to each token overall not per layer. What it means is that if token 1 pays attention to token 0 in layer 1, and token 2 pays attention to token 1 in layer 2, this is counted as attending to token 0. However, these weights are not what is looked at when talking about attention patterns. Attention patterns look at how attention is distributed in a single layer.  As such, I do not believe that the mapping of the obtained results (that are based on the graph model) to

**Other Comments Or Suggestions:**

No comments other than above.

**Other Strengths And Weaknesses:**

No comments other than above.


## update after rebuttal
My concerns remain largely unaddressed. For example:

1) It is possible that understanding the distribution of attention scores over original tokens would be beneficial. However, this paper makes many claims about the relevance of results with the per-layer attention patterns. For example, Table 1 tries to suggest relevance between the theorems and the attention sink phenomenon. These claims are made regularly in the paper and are the essence of how the results are shown to be important. However, these are in no way valid since the thoerems do not apply to per-layer attention patterns at all. As such, I find them extremely misleading to the community.

2) I am still unsure about experimental results showing that the models can not learn positional biases since previous work also include empirical work that show it is possible to decode position information when no explicit positional encoding is provided. I believe a mismatch in settings might be the culprit and I would like to ask the authors to carefully investigate this.

3) Additionally, conclusions drawn from the upper boud remain questionable. For example in the comment "later tokens become progressively less influential, while earlier tokens retain more aggregate attention", the upper bound only shows the first one. However, it does not mean the second one is necessarily true unless a lower bound is drawn. If all the scores go equally to 0, the upper bound would still hold. It might be possible to show such lower bound as well, given that the scores sum to 1 but this needs to be done before making the second part of the claim mentioned above.

My score remains unchanged.

**Questions For Authors:**

See above.

**Relation To Broader Scientific Literature:**

No comments.

**Theoretical Claims:**

Look at the above

---

> ### Author Rebuttal · Authors · 2025-04-01
>
> We thank the reviewer for providing thoughtful feedback. Below, we provide detailed responses to the comments.
>
> > Theorem 4.1 only yields an upper bound on the weight given to a token.
>
> Thank you for the comment. We agree that Thm 4.1 provides an upper bound rather than a strict inequality, and we do not claim it guarantees a strict bias toward earlier tokens in every setting. Rather, it characterizes an emergent tendency, rooted in the causal mask, for earlier tokens to accumulate more attention as the number of attention layers increases.
>
> - Importantly, while the upper bound is not always tight, it is not vacuous: it decays exponentially to $0$ for tokens $i\geq 2$ as attention deepens, meaning it cannot be arbitrarily large. This suggests that—regardless of specific token embeddings—later tokens become progressively less influential, while earlier tokens retain more aggregate attention. One can derive more precise and stronger results by imposing additional assumptions regarding token embeddings or model weights. For example, for sequences of identical tokens, the upper bound becomes exact.
>
> - Moreover, our empirical results support these core insights from Thm 4.1. As shown in Fig. 2, even in the absence of explicit positional bias in the data, models with causal masking exhibit clear early-token bias—an effect that strengthens with depth. This empirical observation closely aligns with our theoretical findings, suggesting that the upper bound meaningfully captures the inductive bias induced by causal masking.
>
> > Attention patterns look at how attention is distributed in a single layer.
>
> Thank you for the comment.
>
> - While attention patterns are often analyzed at the single-layer level, such analyses may miss how information propagates and accumulates across layers—a key aspect of how transformers build contextual representations. A central contribution of our work is to model this global effect of attention across layers. In your example, while token 2 does not directly attend to token 0, it inherits information from token 0 through token 1. Such multi-hop influences are not captured by per-layer attention patterns but are critical for understanding deep model behavior.
>
> - Our graph-theoretic framework is designed precisely to capture these *multi-step dependencies by modeling the global effect of attention across layers*. This complements prior layerwise analyses and aligns with recent work (Abnar et al., 2020; Barbero et al., 2024; Wu et al., 2024) that emphasizes the importance of analyzing multi-layer attention compositionally.
>
> By tracing these cumulative token interactions, our framework offers a complementary and more holistic view of attention dynamics. We hope that our methodology can serve as a useful tool for the community and help guide future investigations into multi-layer attention mechanism.
>
> > In Kazemnejad et. al. the authors explicitly show a set of weights for a transformer model that lead to it learning positional indices whereas the paper is claiming this can not be achieved based on experiments.
>
> Thank you for the comment. We would like to clarify the relationship between our work and the findings of Kazemnejad et al.:
>
> - Kazemnejad et al. provide a theoretical construction showing that positional information can, in principle, be encoded using only causal masking via hand-crafted weights. However, their result does not reflect what typically emerges through standard training.
>
> - Our work builds on this insight by asking a complementary question: Do transformer models actually learn positional encodings in practice when trained without explicit positional signals? We address this through a controlled task where position is critical and observe whether models develop positional awareness under causal masking alone.
>
> - As shown in Sec. 5.2 and Fig. 3, models trained with explicit positional encodings (sin PE or RoPE) capture both start and end-of-sequence biases, while models using only causal masking consistently fail to capture end-of-sequence patterns. This suggests that while position can theoretically be represented via causal masking, *it does not seem to emerge naturally in practice*, even when position is highly informative. Our theory supports this by showing that causal masking induces an inductive bias toward earlier positions.
>
> Thus, rather than contradicting Kazemnejad et al., our work provides a more practical and deeper understanding of how position bias actually manifests in Transformers under different PEs. We will clarify this distinction more explicitly in the final version of the manuscript.
>
>
> We appreciate your questions and comments very much. Please let us know for any further questions.
>
> ---
> References
>
> Abnar et al. Quantifying Attention Flow in Transformers
>
> Wu et al. On the Role of Attention Masks and LayerNorm in Transformers
>
> Barbero et al. Transformers need glasses! Information overSquashing in Language Task

---

### Decision · Program_Chairs · 2025-05-01

**Decision:**

Accept (poster)

**Comment:**

This paper has received a wide range of reviews. On one hand, some reviewers acknowledge the contributions of this paper, including the new theoretical results for position bias in autoregressive language models and new explanations about how model depths affect positional biases. However, on the other hand, reviewers have raised strong concerns about the mismatch between the theory and empirical observations. Specifically, the authors develop a theory for the attention scores over original tokens, while the empirical observations, such as attention sink, are reported in terms of per-layer attention patterns. Additionally, the paper only states a correlation between attention sinks and the developed theoretical results, without providing an underlying explanation for why attention sinks emerge.

Overall, despite the debates and potential issues with this paper, most reviewers have provided positive evaluations and believe that the identified issues can be addressed in the camera-ready version. The authors should seriously consider these critiques and make the necessary modifications accordingly.